# A chemical–genetic interaction map of small molecules using high-throughput imaging in cancer cells

Marco Breinig[1,2,†], Felix A Klein[3,†], Wolfgang Huber[3,*] & Michael Boutros[1,2,**]

## Abstract

Small molecules often affect multiple targets, elicit off-target effects, and induce genotype-specific responses. Chemical genetics, the mapping of the genotype dependence of a small molecule's effects across a broad spectrum of phenotypes can identify novel mechanisms of action. It can also reveal unanticipated effects and could thereby reduce high attrition rates of small molecule development pipelines. Here, we used high-content screening and image analysis to measure effects of 1,280 pharmacologically active compounds on complex phenotypes in isogenic cancer cell lines which harbor activating or inactivating mutations in key oncogenic signaling pathways. Using multiparametric chemical–genetic interaction analysis, we observed phenotypic gene–drug interactions for more than 193 compounds, with many affecting phenotypes other than cell growth. We created a resource termed the Pharmacogenetic Phenome Compendium (PGPC), which enables exploration of drug mode of action, detection of potential off-target effects, and the generation of hypotheses on drug combinations and synergism. For example, we demonstrate that MEK inhibitors amplify the viability effect of the clinically used anti-alcoholism drug disulfiram and show that the EGFR inhibitor tyrphostin AG555 has off-target activity on the proteasome. Taken together, this study demonstrates how combining multiparametric phenotyping in different genetic backgrounds can be used to predict additional mechanisms of action and to reposition clinically used drugs.

**Keywords** compound mode of action; drug synergism; high-content imaging; isogenic cell lines; systems pharmacology

**Subject Categories** Genome-Scale & Integrative Biology; Pharmacology & Drug Discovery

**Mol Syst Biol. (2015) 11: 846**

## Introduction

Target-centered high-throughput screening has been successful for identifying inhibitory small molecules for drug development pipelines. However, many drugs fail at later stages because of unintended side effects or unfavorable toxicological profiles (Lord & Ashworth, 2010; Bowes *et al*, 2012). These failures are a major factor in the high total costs of drug development and increase the societal burden in developing new and effective therapies. While an initial focus on target activity often yields highly effective small molecules, it may lead to lack of depth and resolution in the characterization of potential polypharmacology (Hopkins, 2008). In addition, even in cases when the targets of a drug are well defined, unanticipated dependencies on specific genetic backgrounds can limit its application. This is particularly relevant in cancer, as cancer cells harbor many somatic mutations (Al-Lazikani *et al*, 2012).

To decrease high attrition rates in drug development, it was proposed to perform comprehensive pharmacological profiling early in the drug development process (Bowes *et al*, 2012). Such approaches, also termed "systems pharmacology", attempt to integrate the identification of a drug's mode of action(s), its polypharmacology, and genotype dependencies (Sorger *et al*, 2011). With few exceptions (Bodenmiller *et al*, 2012; Kleinstreuer *et al*, 2014), scalable methods to this end have been rather difficult to implement. In addition to transcriptome profiling (Lamb *et al*, 2006; Iorio *et al*, 2010), phenotypic profiling by cellular imaging has been deployed as a strategy for delineating a compound's mode of action by comparing drug-specific phenotypic responses (Perlman *et al*, 2004; Young *et al*, 2008; Gustafsdottir *et al*, 2013).

Recently, cell-based viability screens have been used to generate large datasets by profiling up to 350 drugs against compendia of cancer cell lines (Barretina *et al*, 2012; Garnett *et al*, 2012; Basu *et al*, 2013). These and other studies (Torrance *et al*, 2001; Muellner *et al*, 2011; Kittanakom *et al*, 2013) identified gene–drug associations that may underlie genotype-dependent resistance and sensitivity. So far, gene–drug interaction analyses were limited by

1 Division of Signaling and Functional Genomics, German Cancer Research Center (DKFZ), Heidelberg, Germany
2 Department of Cell and Molecular Biology, Heidelberg University, Heidelberg, Germany
3 European Molecular Biology Laboratory (EMBL), Genome Biology Unit, Heidelberg, Germany
*Corresponding author. Tel: +49 6221 3878823; E-mail: whuber@embl.de
**Corresponding author. Tel: +49 6221 421950; E-mail: m.boutros@dkfz.de
† These authors contributed equally to this work

their focus on cell proliferation and viability as a phenotypic readout.

In this study, we profiled gene–drug interactions for more than 1,200 pharmacologically active compounds by high-throughput imaging. Multivariate phenotypic responses of cancer cells were measured in different isogenic genetic backgrounds. In total, we measured 300,000 drug–gene–phenotype interactions to create a resource termed the Pharmacogenetic Phenome Compendium (PGPC). This resource unveiled genotype-specific drug responses and predicted drug combinations in cancer cells. Exploring the PGPC, we could see instances of pathway crosstalk, compound mode of action, and off-target effects. We provide access to the pharmacogenetic phenotypes through an interactive webpage (http://dedom-ena.embl.de/PGPC) and as a Bioconductor/R package.

# Results

## Detection of complex phenotypes across multiple drugs and genetic backgrounds

We established a high-throughput method to quantitatively measure genotype-dependent drug effects on cellular phenotypes using automated microscopy and image analysis (Fig 1A). We selected a panel of 12 isogenic knockout cell lines with mutations in key oncogenic signaling pathways (Fig EV1 and Materials and Methods): three HCT116 colon cancer cell lines where the oncogenic mutation of either *CTNNB1* (β-catenin), *KRAS*, or *PI3KCA* (PI3K) was deleted, leaving only the respective wild-type allele, as well as seven knock-out cell lines for *PTEN*, *AKT1*, *AKT1*, and *AKT2* together (*AKT1/2*), *MAP2K1* (*MEK1*), *MAP2K2* (*MEK2*), *TP53*, and *BAX* and two parental HCT116 cell lines (P1 and P2). HCT116 cells were chosen as a model system since multiple well-characterized isogenic derivatives are available (Torrance *et al*, 2001; Chan *et al*, 2002; Samuels *et al*, 2005; Ericson *et al*, 2010), several of which have previously been used for large-scale compound and RNA interference screens (Torrance *et al*, 2001; Vizeacoumar *et al*, 2013).

HCT116 cells of the indicated genotype were seeded in 384-well plates and compound screens were carried out. We chose a drug library with 1,280 pharmacologically active compounds affecting a broad spectrum of cellular processes and major drug target classes (Table EV1) in order to obtain a comprehensive view of phenotypic–pharmacogenetic effects. Briefly, compounds were added at a single concentration of 5 μM and, after 48 h, cells were stained for DNA and actin to monitor nuclear and cellular phenotypes (Fig 1A). Experiments were performed in two biological replicates, and a total of 294,912 images were analyzed. Images that did not pass quality control were excluded from further analysis (Appendix Fig S1). On average, 7,000 cells were analyzed per well. Building upon a previously established automated image analysis pipeline (Pau *et al*, 2010), we extracted, for each well, 385 quantitative phenotypic features of cellular morphology, including cell number as a measure of overall cell proliferation and viability, resulting in more than 14,000,000 measurements. The reproducibility of phenotypic features was high, with 310 features showing a correlation of > 0.7 (Spearman correlation coefficient) between biological replicates (Fig 1B). To select the most informative and non-redundant phenotypic features, we employed a previously established stepwise selection procedure (Laufer *et al*, 2013). This resulted in a set of 20 features (Fig 1C, Appendix Fig S2), which we grouped into five phenotypic categories. We visualized them by radar charts, which we term phenoprints (Fig 1D).

We explored these charts together with the original images to assess their ability to report specific drug-induced morphological changes. Treatment of the parental HCT116 cells with microtubule-targeting compounds caused apoptosis (Fig 1F and G), and topoisomerase inhibitor treatment resulted in increased nuclear and cellular size as compared to DMSO-treated cells (compare Fig 1E with Fig 1H and I), a phenotype attributed to mitotic catastrophe (Maskey *et al*, 2013). Further examples of characteristic phenotypes included cell death with aberrantly shaped nuclei and locally enhanced actin intensity in the few remaining cells (ouabain, Fig 1J) as well as elongated cells (rottlerin, Fig 1K). Overall, phenoprints served as compact visualizations of drug-induced phenotypes. Drugs sharing the same target resulted in similar phenotypes and had similar phenoprints, as demonstrated by microtubule-targeting compounds (Fig 1F and G), and topoisomerase inhibitors (Fig 1H and I).

Together, these results demonstrate that our assay and data analysis pipeline produces quantitative multivariate feature vectors, or phenotypic signatures, that capture compound-induced cellular phenotypes.

## Quantitative analysis of phenotypic chemical–genetic interactions

To set a baseline for chemical–genetic interaction analysis, we first determined the phenotypic signature of each of the 12 cell lines without drug treatment (Fig EV2). For instance, in contrast to parental

---

**Figure 1. Profiling chemical–genetic interactions using image-based cellular phenotyping.**

A    Schematic representation of the experimental approach.
B    Correlation of phenotypic features between replicates. Features are ranked from left to right by the degree of their correlation. 310 features had a Pearson correlation > 0.7.
C    Feature selection by dimensionality reduction. Stepwise selection based on linear decomposition resulted in a set of the 20 most informative and non-redundant phenotypic features.
D    Phenoprinting. 20 selected features were visualized as "phenoprints", that is, phenotypic drug response signatures. Features were grouped into five broad phenotypic categories: cell number, DNA texture/intensity, nuclear shape, cell shape, and actin texture/intensity. Phenoprint of parental HCT116 cells (P1) treated with DMSO (Colors: cyan, DNA; red, actin). Scale bar, 20 μm.
E–K    Drug-induced phenotypic changes are captured by phenoprints. Parental HCT116 cells (P1) treated with bioactive compounds. Microtubule inhibitors induce condensed nuclei and nuclear blebs (indicative of apoptotic cells), and topoisomerase inhibitors increase nuclear and cellular size (indicative of mitotic catastrophe). Ouabain, a Na/K-pump inhibitor, induces cell death with a cellular morphology different from microtubule-targeting agents. Rottlerin, originally classified as a PKC inhibitor, induces cell elongation captured by a distinct phenoprint. Scale bars, 20 μm.

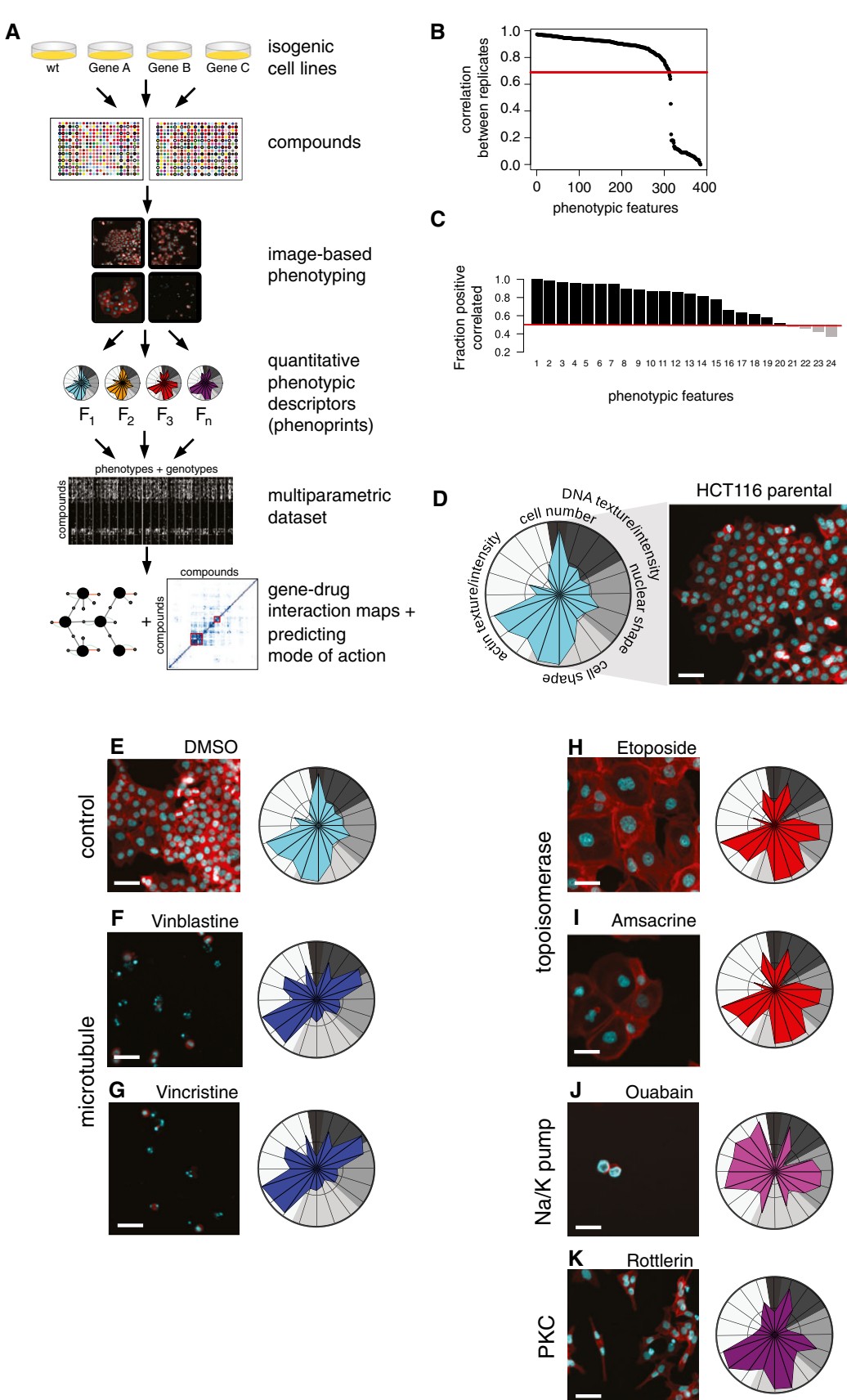

**Figure 1.**

HCT116 cells (*CTNNB1* mutant [mt], (HCT116 $^{CTNNB1\ wt\ +/mt\ +}$)), *CTNNB1* wild-type (wt) cells (HCT116 $^{CTNNB1\ wt\ +/mt\ -}$) showed protrusions of the cell body, a morphology previously associated with a mesenchymal-like phenotype (Caie *et al*, 2010; Sero *et al*, 2015), as well as an irregular nuclear shape (Fig 2A). Then looking at drug-treated cells, we observed genotype-dependent and genotype-independent phenotypes: For example, etoposide increased nuclear and cellular size in the parental and in the *CTNNB1* wt cells, and the phenoprints indicated largely comparable changes in shape. In contrast, the spindle toxin colchicine induced an apoptosis phenotype in parental HCT116 cells, whereas we observed increased sizes for the *CTNNB1* wt cells. Analogously, the histone methyl-transferase inhibitor BIX01294 had a moderate impact on parental HCT116 cells, but led to decreased cell size and altered nuclear shape in *CTNNB1* wt cells (Fig 2A).

Next, we calculated interaction coefficients (Horn *et al*, 2011; Laufer *et al*, 2013) for each of the 1,280 compounds across the 20 features in all genotypes tested. Briefly, the approach accounts for baseline genotype and drug effects with an ANOVA-type approach, and the resulting interaction coefficients measure the difference between the observed phenotype for a given drug–genotype combination and the expected phenotype if the drug and genotype effects were combined independently (Materials and Methods). For instance, a negative interaction coefficient for cell number signified a genotype-specific growth defect. This approach allowed a quantitative determination of a broad spectrum of phenotype-specific interactions. For example, colchicine and BIX01294 treatment revealed interactions for multiple phenotypic features in *CTNNB1* wt cells, whereas we did not observe significant interactions affecting cell number, that is, cell proliferation and viability (FDR < 0.01, Fig 2B and Appendix Fig S3). This indicates that gene–drug interactions for colchicine or BIX01294 were specifically seen in cell morphology phenotypes, while effects on cell number were independent of mutant versus wild-type *CTNNB1* genotype.

Our analysis yielded a dataset, termed the Pharmacogenetic Phenome Compendium (PGPC), comprising information on more than 300,000 drug–gene–phenotype interactions. Across all 20 phenotypic features investigated, a total of 2,359 significant chemical–genetic interactions were observed (0.8% of all possible interactions; FDR < 0.01). These interactions were associated with 193 compounds (15.1% of compounds tested; Appendix Fig S4). The majority of chemical–genetic interactions did not significantly affect cell growth. For example, 204 chemical–genetic interactions were exclusively due to phenotypic features associated with nuclear shape, whereas only 16 interactions were based on an analysis of cell number (Fig 2C). Only 14 compounds (1.1% of compounds tested) revealed significant interactions for cell number (Appendix Fig S4). Together, these results show that our multiparametric approach provided increased coverage and sensitivity for gene–drug interaction mapping.

Many compounds specifically interacted with few genotypes; for instance, 90 of the 193 compounds had interactions with a single genotype (Fig 2D). We also noted a trend toward higher number of interactions involving cell lines in which the genotype itself had a pronounced phenotypic effect, including cell number (e.g., *CTNNB1* wt cells; Figs 2E and EV2 and Appendix Fig S5A and B). These findings are reminiscent of results reported for genetic interactions in yeast, where stronger effects of single gene deletions correlated with a higher number of interactions (Costanzo *et al*, 2010). When focusing on specific genotypes, we observed that *MEK1* KO cells presented more interactions compared to *MEK2* KO (Fig 2E). Possible reasons for this observation include different levels of expression of MEK1 and MEK2, and some degree of functional specialization between MEK1 and MEK2 (Catalanotti *et al*, 2009; Scholl *et al*, 2009). We further examined the similarity of cell lines by unsupervised clustering of their interaction profiles and found that *KRAS* wt (HCT116 $^{KRAS\ wt\ +/mt\ -}$) and *MEK1* KO cells grouped together (Appendix Fig S5C). This finding is in agreement with a report demonstrating that MEK1 and not MEK2 acts as the crucial modulator in the RAS/MAPK signaling branch (Catalanotti *et al*, 2009).

We also observed more interactions for *AKT1/2* double KO cells compared with *AKT1* KO alone (Fig 2E). This is likely due to functional redundancy, consistent with studies that demonstrated that neither *AKT1* nor *AKT2* KO affected cell growth in HCT116 cells, whereas simultaneous *AKT1/2* KO reduced proliferation and impaired metastasis formation (Ericson *et al*, 2010). Together, these results demonstrate that quantitative multi-phenotype chemical–genetic interaction profiles convey biologically relevant information.

### A phenotypic chemical–genetic interaction map

To obtain a global overview of associations between signaling pathway states, related genes, and phenotypic drug effects, we created a

---

**Figure 2. Quantitative analysis of phenotypic chemical–genetic interactions.**

A  Drugs induce either convergent or divergent phenotypic alterations depending on genetic backgrounds as revealed by visual inspection. Phenotypes for parental HCT116 cells (P1; *CTNNB1* mutant (mut); HCT116 $^{CTNNB1\ wt\ +/mt\ +}$) and *CTNNB1* wild-type (wt) (HCT116 $^{CTNNB1\ wt\ +/mt\ -}$) cells, that is, HCT116 cells with a knockout of the mutant allele, differ under control conditions (DMSO). Treatment with etoposide induces an increase in nuclear and cell size in both genetic backgrounds. Colchicine induces apoptosis in parental HCT116 cells and an increase in nuclear and cell size in *CTNNB1* wt (HCT116 $^{CTNNB1\ wt\ +/mt\ -}$) cells. BIX01294 moderately affects phenotypic features in parental cells, but induces cell condensation in *CTNNB1* wt (HCT116 $^{CTNNB1\ wt\ +/mt\ -}$) cells. Colchicine and BIX01294 reduce cell number independent of genotype. Colors: cyan, DNA; red, actin. Scale bars, 20 μm.

B  Quantitative analysis of chemical–genetic interactions across multiple phenotypic features. Chemical–genetic interactions were calculated for all 20 phenotypic features as described. Colchicine and BIX01294 display multiple interactions in *CTNNB1* wt (HCT116 $^{CTNNB1\ wt\ +/mt\ -}$) cells. Interactions are scaled to range of 0 to 1. *FDR < 0.01, highlighted in red.

C  Overlap of chemical–genetic interactions between phenotypic categories. Zero values have been omitted for better readability.

D  Specificity and pleiotropy of gene–drug interactions. The fraction of genetic backgrounds is shown for which compounds reveal at least one significant interaction (FDR < 0.01).

E  Number of interactions per genetic backgrounds. Different genotypes reveal varying numbers of interactions across the 20 phenotypic features investigated (FDR < 0.01).

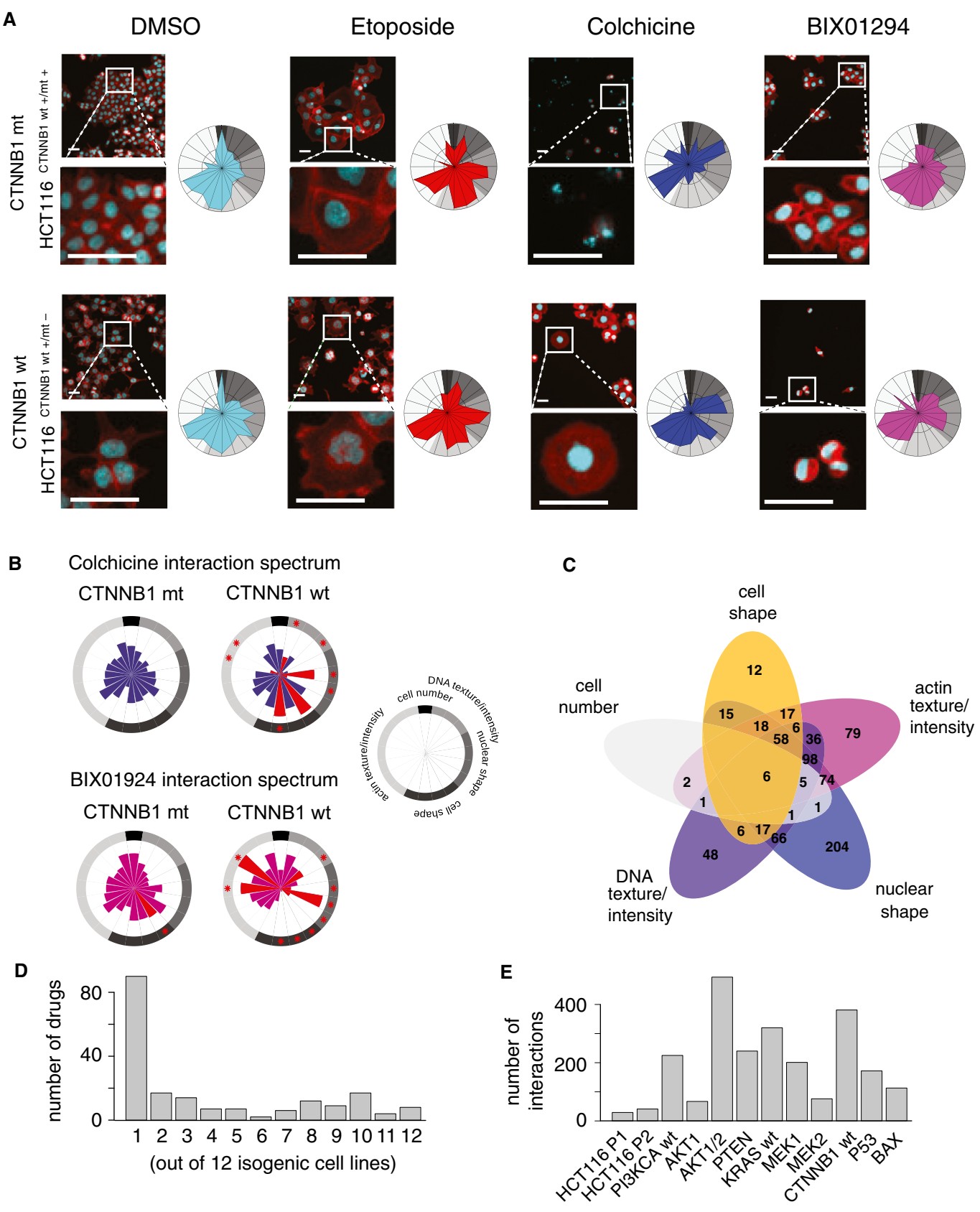

**Figure 2.**

map of gene–drug interactions (Fig 3). Among others, we observed interactions between the PI3K inhibitor wortmannin and *KRAS* wt (HCT116 $^{KRAS\ wt\ +/mt\ -}$) cells, suggesting a higher dependence on PI3K signaling of *KRAS* wt cells as compared to *KRAS* mt parental HCT116 cells (HCT116 $^{KRAS\ wt\ +/mt\ +}$). Interactions between KRAS and PI3K have previously been reported in HCT116 cells (Torrance *et al*, 2001; Vizeacoumar *et al*, 2013). Moreover, we discovered interactions between two casein kinase 2 (CK2) inhibitors and *PI3KCA* wt (HCT116 $^{PI3KCA\ wt\ +/mt\ -}$), but not *PTEN* KO and *AKT1/2* KO cells. Mechanistically, CK2 has been demonstrated to inhibit PTEN and directly activate AKT; both events trigger PI3K/AKT pathway activity (Torres & Pulido, 2001; Di Maira *et al*, 2005). Our findings are consistent with these observations and suggest that CK2 does not compensate perturbations at the hierarchical level of AKT, but can compensate perturbations at the level of PI3K and

potentially also upstream of PI3K, that is, at the level of receptor tyrosine kinases. The latter hypothesis is in agreement with a recent finding that demonstrated synergism for the combinatorial pharmaceutical inhibition of CK2 and EGFR (Bliesath *et al*, 2012). Our map further revealed interactions between a MNK1 inhibitor and both *PI3KCA* wt (HCT116 $^{PI3KCA\ wt\ +/mt\ -}$) and *KRAS* wt (HCT116 $^{KRAS\ wt\ +/mt\ -}$) cells, suggesting that RAS/MAPK and PI3K signaling converge at the level of this kinase. MNK1 is a downstream factor of RAS/MAPK signaling involved in translational control via regulation of eIF4E (Wang *et al*, 2007). In support of this interpretation, a link between PI3K/mTOR and MAPK/MNK1 signaling to eIF4E phosphorylation has been demonstrated (Wang *et al*, 2007).

Together, these results indicate that phenotypic chemical–genetic interaction maps can be used to investigate crosstalk between

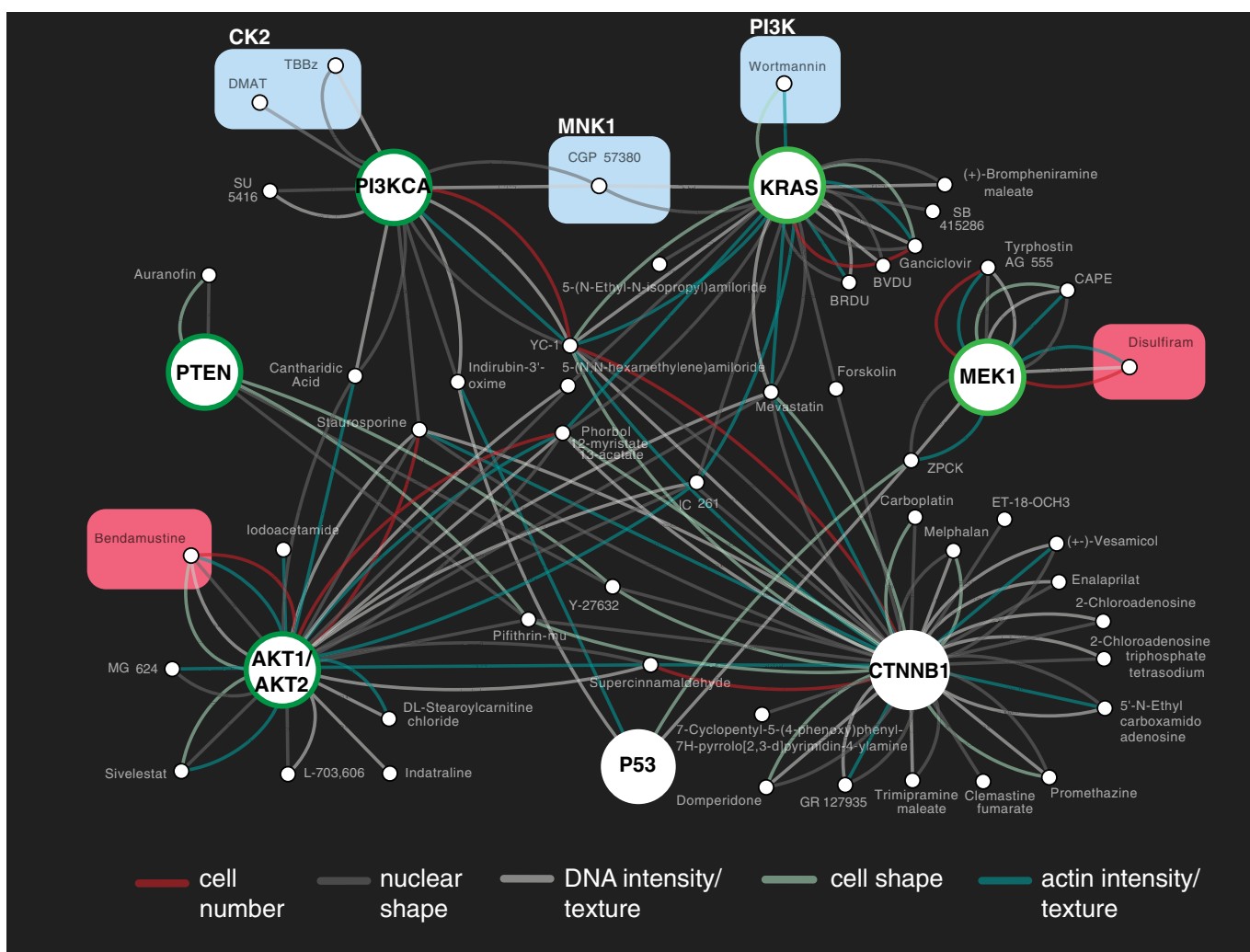

**Figure 3.　Multiparametric chemical–genetic interaction map of colon cancer cells.**
Large nodes represent genetic backgrounds. CTNNB1: HCT116 $^{CTNNB1\ wt\ +/mt\ -}$, KRAS: HCT116 $^{KRAS\ wt\ +/mt\ -}$, MEK1: HCT116 $^{MAP2K1\ -/-}$, PI3KCA HCT116 $^{PI3KCAwt\ +/mt\ -}$, PTEN: HCT116 $^{PTEN\ -/-}$, AKT1/AKT2: HCT116 $^{AKT1\ -/-;\ AKT2-/-}$, P53: HCT116 $^{TP53\ -/-}$. Node border color indicates association with either the PI3K (green) or the MAPK (light green) signaling pathway. Small nodes represent compounds. An edge indicates that a drug–gene interaction was observed for at least 1 of the 5 phenotypic categories. For display, the network was filtered to show only compounds interacting with at most 3 of 12 genetic backgrounds and affecting more than 1 of 20 phenotypic features (FDR < 0.01). Compounds are highlighted for which a gene–drug interaction was used to predict links between different signaling pathways (cyan), or to extrapolate drug–gene interactions to drug–drug combinations (red).

signaling pathways and to derive insights how drugs perturb genetic networks.

## Extrapolating drug–gene interactions to drug–drug combinations

We next asked whether the drug–gene interaction map could be used to predict effective drug–drug combinations. We reasoned that if a drug's effect is particularly strong in a genetic background with diminished activity of a certain pathway or process, such an effect might also be achieved in other backgrounds by use of a relevant chemical inhibitor. Focusing on drug effects affecting cell growth, we first set our attention on bendamustine, a DNA-alkylating agent approved for the treatment of chronic lymphocytic leukemia (CLL) that does not have cross-resistance with standard alkylating agents (Keating *et al*, 2008). Bendamustine markedly reduced cell number specifically in *AKT1/2* double KO cells (Fig 4A, Appendix Figs S6 and S7A). We therefore tested the combination of bendamustine with small molecule AKT inhibitors and observed that while AKT inhibition alone had a negligible effect on viability of the parental HCT116 cells, in combination with bendamustine the effect was significantly stronger than the bendamustine single-agent treatment (Fig 4B). This synergy was specific for AKT inhibition, as the combination of bendamustine with MEK inhibitors did not elicit stronger effects compared with either single-agent treatment (Fig EV3A and B).

A second instance involved the aldehyde dehydrogenase inhibitor disulfiram, a drug used for the treatment of alcoholism (Lövborg *et al*, 2006). Disulfiram showed a synthetic lethal interaction with *MEK1* KO cells (Fig 4C, Appendix Figs S6 and S7B). Therefore, we hypothesized that an analogous synergy could exist between disulfiram and MEK inhibitors. The combination of disulfiram and MEK inhibitors resulted in significantly stronger viability reduction compared to disulfiram alone, while MEK inhibitors alone had no detectable effect (Fig 4D). Both drug combinations (bendamustine with AKT inhibitors, and disulfiram with MEK inhibitors) were also effective in a second colon cancer cell line (DLD-1; Fig EV3C–H).

Overall, these results demonstrate that such chemical–genetic resources can be used to derive specific predictions of synergism for drug combinations.

## Phenotype-based similarity clustering of drug interaction profiles aids compound characterization

Next, we analyzed the similarity of interaction profiles by unsupervised clustering, using one minus the correlation coefficient of drug profiles as a measure of dissimilarity. This analysis revealed several tight clusters of drugs with known shared mode of action (Fig 5A and Appendix Fig S8). To further dissect these results, we visualized the chemical similarities between compounds by their Tanimoto distances (Fig 5A, in orange). This showed that our phenotypic interaction data-driven clusters could in some cases be explained by chemical similarity, whereas in other cases, chemically distinct compounds shared high interaction profile similarity (see Table EV2 for detailed information related to highlighted clusters C1–C18 and compounds).

For example, cluster C1 contained compounds that target tubulin, including taxol, vincristine, and nocodazole; these compounds do not share a high degree of chemical similarity. Similarly, the MEK

inhibitors PD98059 and U0126 are not highly structurally related but clustered tightly (C2, including a spike-in control). We also found clusters with structural and functional similarity (C3–C5). Examples include the p38 inhibitors PD169316 and SB202190 (C3), and the synthetic glucocorticoids betamethasone and beclomethasone (C4). There were also instances of structurally highly related compounds with different bioactivities (arrows, off-diagonal). For example, the two cytosine analogues 5-azacytidine and ara-c did not directly cluster together (arrow 1), consistent with the fact that only 5-azacytidine inhibits DNA methyltransferases (Christman, 2002).

The map also included divergent associations for compounds that are annotated to interfere with the same biological process (C6–C10). For example, we observed a cluster including two DNA-alkylating agents (C9), whereas the DNA-alkylating agent bendamustine (C10) did not reveal a similar profile and instead shared a signature with pifithrin-μ, which interferes with p53 (Strom *et al*, 2006). Our observation is in accordance with a previously suggested different mode of action of bendamustine compared with other DNA-alkylating agents (Leoni *et al*, 2008).

The map contained several instances of clusters of compounds that target different effector molecules in connected biological processes (C11–C16). For example, C16 included inhibitors of folate metabolism, the DNA methyltransferase inhibitor 5-azacytidine, and the iron chelator phenanthroline. Associations between iron and folate metabolism have been reported (Oppenheim *et al*, 2000), and folate metabolism is linked to DNA methylation (Crider *et al*, 2012).

The map further allowed us to infer primary mode of action and off-target effects of compounds (C17, C18 see below). For example, C17 clustered the anti-helmitic compound niclosamide, which uncouples oxidative phosphorylation (Weinbach & Garbus, 1969; MacDonald *et al*, 2006), and rottlerin, a compound thought to target PKC. This result is in accordance with the finding that rottlerin uncouples oxidative phosphorylation and supports the view that the classification of rottlerin as a PKCδ inhibitor is incorrect (Soltoff, 2001).

While in some instances the correlation of drug profiles was a result of coordinated subtle covariation across multiple cell lines and phenotypes, in other cases it appeared to be driven by the similarity of distinctive phenotypes of individual isogenic cell lines. For example, and as already introduced above, the CK2 inhibitors DMAT and TBBz (Fig 5A, C5) affected nuclear shape features specifically in cells with only the wild-type copy of PI3KCA (HCT116 $^{\text{PI3KCA wt +/mt −}}$) (Appendix Fig S9 and Fig 3), whereas the compounds ARP101 and YC-1 in C15 affected distinct phenotypic features in cells with only the wild-type copies of *CTNNB1* (HCT116 $^{\text{CTNNB1 wt +/mt −}}$) and *PI3KCA* (HCT116 $^{\text{PI3KCA wt +/mt −}}$) (Appendix Fig S9). Further, compounds in C18 had distinctive interactions with *MEK1* KO cells (HCT116 $^{\text{MAP2K1 −/−}}$) (Appendix Fig S9 and Fig 3).

Together, these findings demonstrate that phenotypic chemical–genetic interaction profiles provide a rich resource for the characterization of compounds with a broad spectrum of bioactivities.

## Integrating pharmacogenetic and phenotypic information increases sensitivity

We asked to what extent there is a benefit from using multiparametric phenotyping on multiple genetic backgrounds compared to either approach alone, that is, using multiparametric phenotyping of drugs

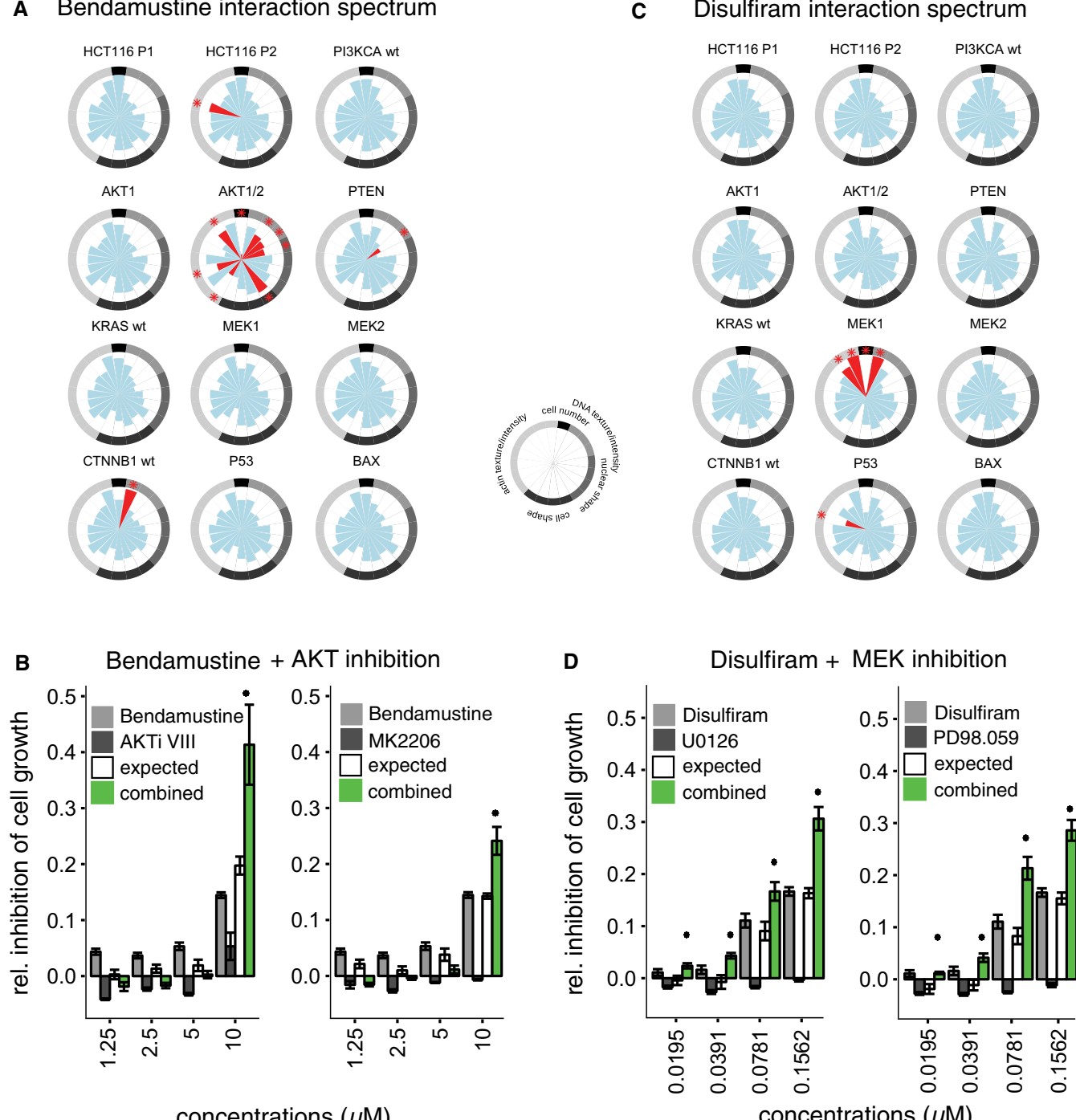

**Figure 4. Extrapolating drug–gene interactions to drug–drug combinations.**

A Bendamustine interaction spectrum. Bendamustine revealed chemical–genetic interactions across multiple phenotypic features including reduced cell number, specifically in *AKT1/2* double KO (HCT116 $^{AKT1\ -/-;\ AKT2\ -/-}$) cells. Interactions are scaled from 0 to 1. *FDR < 0.01, highlighted in red.

B The combination of bendamustine with Akt inhibitors (AKTi VIII or MK2206) resulted in significantly stronger viability reduction compared with either drug alone.

C Disulfiram interaction spectrum. Disulfiram revealed chemical–genetic interactions across multiple phenotypic features including reduced cell number, specifically in *MEK1* KO (HCT116 $^{MAP2K1\ -/-}$) cells. Interactions are scaled from 0 to 1. *FDR < 0.01, highlighted in red.

D The combination of disulfiram and MEK1/2 inhibitors (U0126 or PD98.059) resulted in significantly stronger viability reduction compared with either drug alone.

Data information: Genotypes: HCT116 P1 and P2: HCT116 $^{CTNNB1\ wt\ +/mt\ +;\ KRAS\ wt\ +/mt\ +;\ PI3KCA\ wt\ +/mt\ +}$; PI3KCA wt: HCT116 $^{PI3KCA\ wt\ +/mt\ -}$; *AKT1*: HCT116 $^{AKT1\ -/-}$; *AKT1/2*: HCT116 $^{AKT1\ -/-;\ AKT2\ -/-}$; *PTEN*: HCT116 $^{PTEN\ -/-}$; KRAS wt: HCT116 $^{KRAS\ wt\ +/mt\ -}$; *MEK1*: HCT116 $^{MAP2K1\ -/-}$; *MEK2*: HCT116 $^{MAP2K2\ -/-}$; CTNNB1 wt: HCT116 $^{CTNNB1\ wt\ +/mt\ -}$; *P53*: HCT116 $^{TP53\ -/-}$; *BAX*: HCT116 $^{BAX\ -/-}$. Error bars, means ± s.e.m. $n \geq 3$ of at least three independent experiments that determined cell viability using the CellTiterGlo assay. BI significance is shown, *$P < 0.05$.

on a single cell line or drug–gene interactions using only a viability phenotype, that is, cell number. We evaluated the distribution of correlation coefficients of all drug interaction profiles for each of these alternatives. The integrated approach—combining multiparametric phenotypes with pharmacogenetic interaction mapping—provided better resolution for compound classification compared to either individual approach (Appendix Fig S10). There were instances in which drug–gene interactions using only a viability phenotype, grouped together related compounds, including microtubule inhibitors such as vinblastine and nocodazole (Appendix Fig S10, genotypes). Likewise, there were instances in which phenotypic profiling of drugs using solely the parental HCT116 cell line grouped together related compounds, including the MEK inhibitors PD98059 and U0126 (Appendix Fig S10, multiparametric phenotypes). However, associations between other compounds, including the structurally highly related glucocorticoids betamethasone and beclomethasone (Fig 5A, C4), were only revealed by the combined approach (Appendix Fig S10, genotypes and multiparametric phenotypes).

To measure assay performance, we assembled prior information about target selectivity for the pharmacologically active compounds in the screened library (Table EV1) and grouped the 193 compounds that had interactions into the classes "shared selectivity" or "no shared selectivity". Then, we assessed how well the different approaches separated these classes (Materials and Methods). Integration of phenotypic profiling and pharmacogenetic analyses was superior in predicting shared target selectivity than either individual approach (Fig 5B and C). We further used information about chemical similarity as determined by Tanimoto distances to group drugs into the classes "shared chemical structure" or "different chemical structure" and analyzed how well the different approaches separated these classes (Materials and Methods). Again, the combined strategy revealed better performance (Appendix Fig S11).

Collectively, these results underline the added value of integrating multiparametric phenotypic profiling and pharmacogenetic interaction mapping for compound classification.

### Off-target activity of the EGFR inhibitor tyrphostin AG555

The phenotypic chemical–genetic matrix suggested unexpected relationships that we next confirmed by additional experiments. For example, we observed surprisingly similar interaction profiles for the ALDH inhibitor disulfiram, the EGFR inhibitor tyrphostin AG555, the chymotrypsin inhibitor ZPCK, and the NF-κB inhibitor CAPE (Fig 5A C18 and Fig 6A). Mechanistically, NF-κB activity is known to be tightly regulated by proteasomal degradation of IκB

and the proteasome's chymotrypsin-like activity (Kisselev *et al*, 2012) might explain the functional similarity of ZPCK and CAPE. Additionally, disulfiram was previously shown to impair proteasome function (Lövborg *et al*, 2006). Therefore, we tested whether all compounds in cluster C18, including the EGFR inhibitor tyrphostin AG555, might indeed affect proteasome activity.

To this end, we measured the chymotrypsin-like, trypsin-like, and caspase-like proteasome activities in HCT116 cells treated with disulfiram, ZPCK, tyrphostin AG555, or CAPE. As shown in Fig 6B, these experiments showed significant inhibition of proteasome activities by all four compounds, although weaker than the effect induced by the well-established proteasome inhibitors MG132 and bortezomib. Moreover, we observed that tyrphostin AG555 increased the abundance of ubiquitinylated proteins (Appendix Fig S12). In contrast, two structurally different EGFR inhibitors that were included in our screened compound library, AG1478 and DAPH, did not impair proteasome function, suggesting that the effects of compounds in cluster C18 were not mediated by EGFR inhibition (Fig 6B). These findings identify proteasome inhibition as a previously unrecognized off-target activity of the EGFR inhibitor tyrphostin AG555.

## Discussion

How to integrate the search for off-target activities and genotype-specific effects early in drug development pipelines has remained a largely unresolved issue. Here, we established a chemical–genetic interaction and phenotypic profiling approach to provide multiple layers of quantitative information based on an integrated and standardized screening procedure. Quantitative mapping of chemical–genetic interactions across complex phenotypes improves compound classification and we demonstrate that a rich set of information can be obtained by monitoring phenotypes beyond cell viability, as interactions observed for cell proliferation are only one dimension of the wider range of nonlinear combinatorial effects between compounds and genetic variants. Our experiments reveal the advantages and prospects of multiparametric interaction analyses both at large scale and for specific examples. The methods described here can be adapted and applied to query phenotypic–pharmacogenetic interactions in additional cell lines from various lineages. Finally, we provide a searchable resource for 1,280 pharmacologically active compounds that can serve as a template for larger sets of molecule collections.

Drug development suffers from high attrition rates as critical information about a drug's mode of action and off-target effects is

---

**Figure 5.  Clustering of phenotypic chemical–genetic interactions.**

A   Unsupervised clustering based on the correlation of compound interaction profiles of all 12 genetic backgrounds for 20 phenotypic features (upper left; similarity of multiparametric interaction profiles; color scale: white to blue). Structural similarity of compounds (lower right; color scale: white to orange). Color coding of cluster tree visualizes automated cluster analysis using a 0.6 height cutoff for the cluster tree and the inclusion of > 2 and < 10 drugs per cluster. See text and Table EV2 for details.

B   Interaction profile correlation from the integrated approach is better at predicting compounds' shared target selectivity, as indicated by the shift of the empirical cumulative density functions (ECDF) for shared targets (red curve) compared to non-shared targets (blue curve) in the three panels: Genotypes: data only on cell number using 12 genetic backgrounds; Multiparametric phenotypes: data using image-based 20 phenotypic features of one genetic background (parental HCT116; P1); Genotypes and multiparametric phenotypes: interaction profiles derived from both 12 genetic backgrounds and image-based 20 phenotypic features.

C   Resolution index, ΔAUC, displays the performance by which each strategy separated drugs that share/do not share target selectivity (see text and Materials and Methods for details).

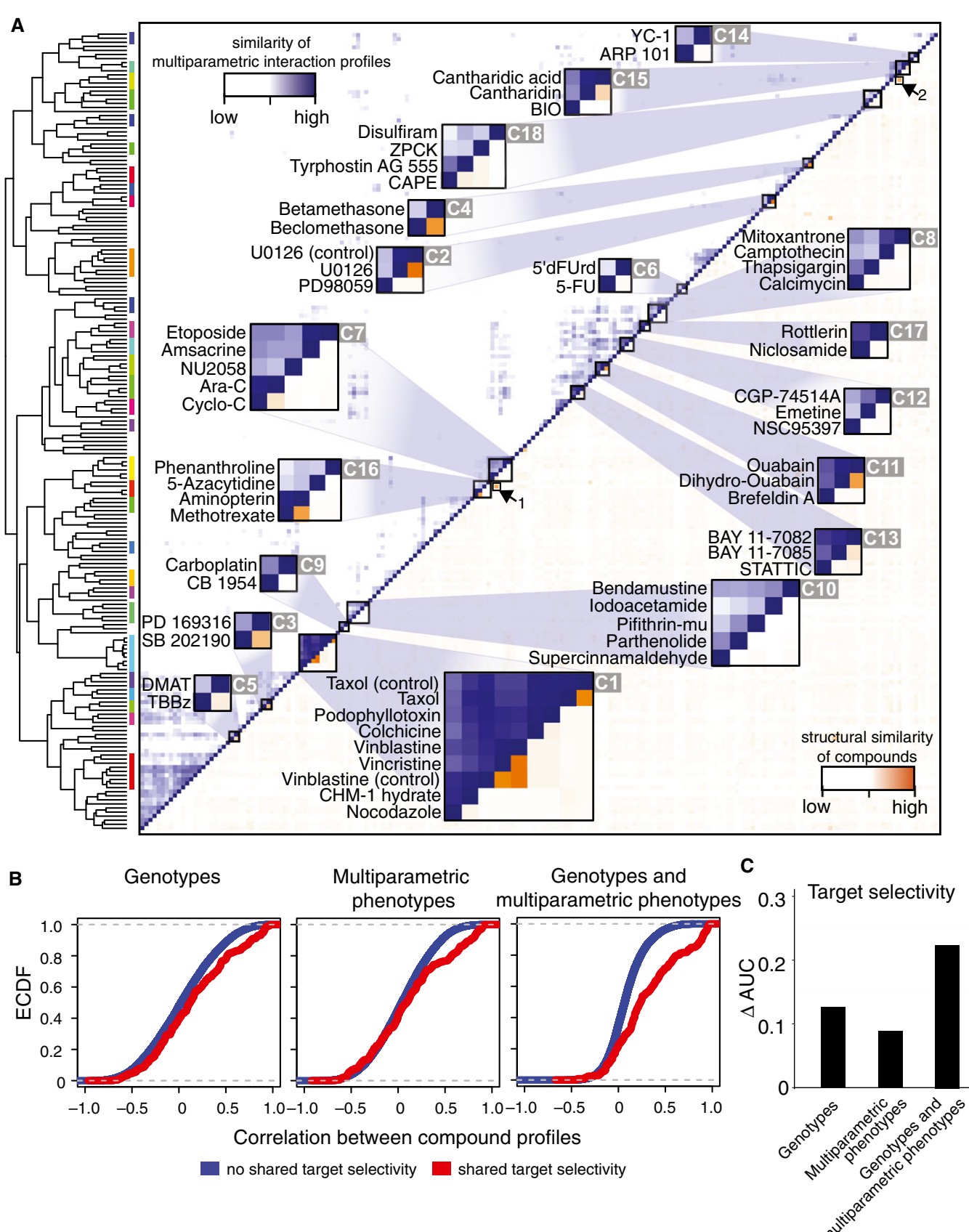

**Figure 5.**

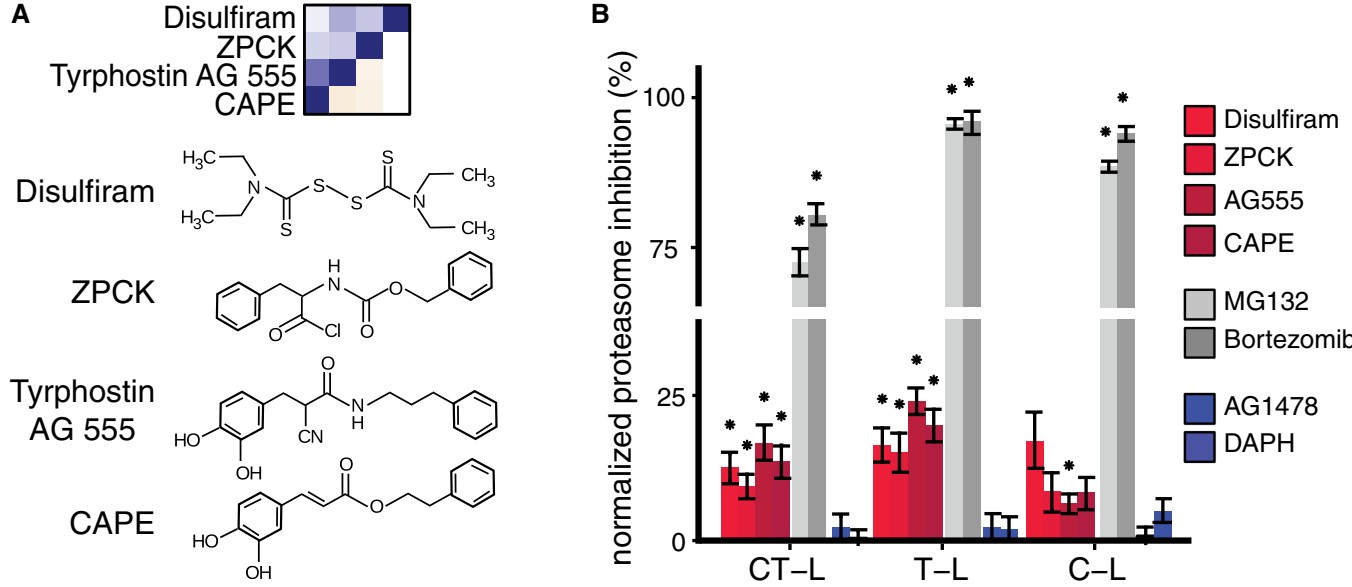

**Figure 6. The EGFR inhibitor tyrphostin AG555 off-targets proteasome function.**

A   Chemical structures for compounds in cluster C18.

B   The EGFR inhibitor tyrphostin AG555 impairs proteasome function. Chymotrypsin-like (CT-L), trypsin-like (T-L), and caspase-like (C-L) activities were measured in HCT116 cells 24 h after addition of the indicated compounds observed in cluster C18 at a concentration of 5 μM. The proteasome inhibitors MG132 and bortezomib served as positive controls. The EGFR inhibitors AG1478 and DAPH are shown as additional controls. Proteasome activity was normalized to DMSO control and corrected for cell viability effects measured using CellTiterGlo. DMSO control determines 0% normalized proteasome inhibition (NPI). Error bars, means $\pm$ s.e.m. $n \geq 5$ of five independent experiments. Asterisk (*) indicates $P < 0.05$ in one sample one-sided $t$-test against NPI = 0.

often learned too late. Comprehensive drug characterization early on in the development process should reduce expensive failure of drugs in late stages. New information-rich, robust, and scalable screening approaches should simultaneously provide information about drug mode of action and polypharmacology, off-target effects, and how drugs affect complex cellular networks by gene–drug interactions (Sorger *et al*, 2011). Interaction maps of compounds as described here could fill the gap between target-oriented small molecule design and the characterization of polypharmacological drugs and might provide important information at crucial decision points.

The resource we describe here enabled us to map gene–drug interactions associated with cancer-relevant pathway activation states and cellular networks. Using interaction network analysis pioneered in model organisms (Costanzo *et al*, 2010), we were able to predict synergism for specific combinations of drugs on the basis of genetic knowledge. For such analysis, the use of genetically engineered isogenic cell lines can be advantageous over cancer cell line compendia. First, drug sensitivity or resistance can be correlated with multiple features co-occurring in the same cell line (Garnett *et al*, 2012), making it challenging to directly identify causal gene–drug interactions. A second point is made by our discovery of the synthetic lethal interaction between the alkylating agent bendamustine and the double *AKT1/2* KO genetic background, which is unlikely to occur naturally in cancer cell lines or primary cells. Based on this finding, we were able to predict and validate synergism of bendamustine and AKT inhibitors in colon cancer cells, shortcutting a potentially more complex combinatorial drug screen. Bendamustine has clinical activity in numerous cancer types (Keating *et al*, 2008)

and AKT inhibitors have recently entered clinical trials including a study investigating the combination of bendamustine with MK2206 and rituximab in CLL patients (NCT01369849). Based on the hypothesis that bendamustine and AKT inhibition affect cancer growth via different mechanisms, synergism between bendamustine and MK2206 has recently been tested and demonstrated in patient-derived CLL cells (Ding *et al*, 2013). Our results provide an unbiased foundation for the combination of these agents in colon cancer cells. We further used our gene–drug interaction data to demonstrate the efficacy of combining disulfiram with MEK inhibitors. Disulfiram is well established for the treatment of alcoholism, and in conjunction with previous findings (Lövborg *et al*, 2006), our data suggest that disulfiram's proteasome inhibitory capacity could be repurposed for anticancer treatment. We further note that the compounds we observed to impair proteasome function (Fig 5A C18 and Fig 6A) primarily interacted with *MEK1* KO cells (Fig 3), which might indicate that the combination of MEK inhibitors with compounds that impair proteasome activity could generally be effective.

Although we initially focused the analysis of drug synergism on the simple phenotypic effect of compounds on cell growth, we anticipate that the integrated phenotypic–pharmacogenetic approach could in the future also be used to predict drug combinations effective on other processes relevant to cancer. More research is needed for a fair assessment of prediction performance, since parameters such as prediction sensitivity and specificity need to be calibrated depending on a drug's single-agent activity, polypharmacology, and its interaction "promiscuity" (Cokol *et al*, 2011).

Our data provide evidence that the integration of phenotypic profiling, which allows monitoring of diverse biological processes including shape changes, mitosis, or apoptosis (Perlman *et al*, 2004; Young *et al*, 2008; Fuchs *et al*, 2010), and pharmacogenetic interaction mapping using isogenic cell lines can characterize drug mode of action and off-target effects at high resolution. Using "guilt-by-association" approaches, which have successfully been employed to infer mechanistic insights for drug action (Perlman *et al*, 2004; Lamb *et al*, 2006; Parsons *et al*, 2006) or to map connected biological processes (Costanzo *et al*, 2010), we predicted associations between drugs and biological pathways. By the same approach, we identified a previously unrecognized off-target effect for the EGFR inhibitor tyrphostin AG555 and demonstrated that this compound impairs proteasome function.

Several extensions of the phenotypic–pharmacogenetic screening assay presented here will be desirable. For example, the resolution of the data would benefit from using multiple doses of each drug (Perlman *et al*, 2004), and their scope could be extended by including larger compound libraries with novel targets. Likewise, a broader set of genetic backgrounds could be used, an aim that now appears quite tractable with isogenic cell lines through use of CRISRP/Cas9 technology (Sander & Joung, 2014). Moreover, the use of additional markers of cellular components for high-content imaging could further increase the phenotypic search space for interactions relevant to an even broader range of biological processes. For example, Gustafsdottir *et al* developed a multiplexing protocol that allows for the detection of seven distinct cell components using six stains and imaging five channels (Gustafsdottir *et al*, 2013).

The results of this study are provided as a resource we termed Pharmacogenetic Phenome Compendium (PGPC). The PGPC has been built using standardized, scalable experimental, and computational methods. The PGPC enables users to search for connected biological processes perturbed by drugs, to investigate pathway crosstalk, and to identify genotype-specific drug responses. It can further be used to predict unexpected effects of drug combinations, compound mode of action, and potential off-target effects. The PGPC is provided for download *in toto* as a data package from www.bioconductor.org, including all raw data and analyses. The entirety of pharmacogenetic phenotypes observed in our experiment can be examined by querying the original images via an accompanying database (http://dedomena.embl.de/PGPC).

Overall, we expect that a systematic survey of relationships between pharmacology, phenotype, and genotype and the integration of emerging pharmacogenetic and phenotypic resources (Young *et al*, 2008; Barretina *et al*, 2012; Garnett *et al*, 2012; Basu *et al*, 2013; Kleinstreuer *et al*, 2014) and complementary strategies such as transcription profiling as a means to infer drug mode of action (Lamb *et al*, 2006) will accelerate the development of quantitative systems pharmacology to deliver efficient genotype-stratified therapeutics and better understanding of side effects.

# Materials and Methods

### Cell lines and cell culture

Parental HCT116 cells (HCT116, P1) were obtained from ATCC. The second parental HCT116 cell line (HCT116, P2) and all isogenic HCT116 cell lines were obtained from Horizon Discovery Ltd. The isogenic cell lines comprised following genotypes: parental HCT116 cell lines (P1 and P2, HCT116 $^{CTNNB1\ wt\ +/mt\ +}$; KRAS wt +/mt +; PI3KCA wt +/mt +); CTNNB1 wt where the oncogenic mutation of *CTNNB1* (β-catenin) was deleted leaving only the respective wild-type allele (HCT116 $^{CTNNB1\ wt\ +/mt\ -}$); *KRAS* wt where the oncogenic mutation of *KRAS* was deleted leaving only the respective wild-type allele (HCT116 $^{KRAS\ wt\ +/mt\ -}$); *PI3KCA* wt where the oncogenic mutation of *PI3KCA* was deleted leaving only the respective wild-type allele (HCT116 $^{PI3KCA\ wt\ +/mt\ -}$); *PTEN* KO (HCT116 $^{PTEN\ -/-}$); *AKT1* KO (HCT116 $^{AKT1\ -/-}$); *AKT1* KO and *AKT2* KO (*AKT1/2* KO, HCT116 $^{AKT1\ -/-;\ AKT2\ -/-}$); *MAP2K1* KO (*MEK1* KO, HCT116 $^{MAP2K1\ -/-}$), *MAP2K2* KO (*MEK2* KO, HCT116 $^{MAP2K2\ -/-}$), *TP53* KO (*P53* KO, HCT116 $^{TP53\ -/-}$); and *BAX* KO (HCT116 $^{BAX\ -/-}$).

All cell lines were authenticated using SNP profiling (Multiplexion). HCT116 cells were propagated in McCoy's 5a modified medium (Life Technologies) supplemented with 10% FBS (Biochrom) and 1% penicillin/streptomycin (P/S) at 37°C and 5% $CO_2$. Sub-cultivation was performed every 4 days at a ratio of 1:10 – 1:20. DLD-1 cells were obtained from ATCC and propagated in Dulbecco's modified Eagle medium (DMEM) (Life Technologies) supplemented with 10% FBS (Biochrom) and 1% P/S at 37°C and 5% $CO_2$. Sub-cultivation was performed every 4 days at a ratio of 1:10 – 1:20.

### Compound treatment

Prior to screening, we prepared serial dilutions of the LOPAC compound library (Sigma) in RPMI medium (Life Technologies) to provide a final stock concentration of 50 μM. Taxol/paclitaxel, vinblastine, and U0126, as well as DMSO (all from Sigma) were included as additional spike-in controls present on all plates. A list of all compounds included in this library is provided with the R/Bioconductor package PGPC and Table EV1. We seeded 1,250 cells in 45 μl McCoy's medium into each well of 384-well clear-bottom microscopy plates (BD Biosciences) and incubated for 1 day at 37°C. 5 μl of compound solution was added using a Beckman Biomek FX robot with 384-well tip head to yield a final concentration of 5 μM and 0.1% DMSO. Cells were cultured for 2 days at 37°C before analysis. For screening, a single drug concentration of 5 μM was used.

### Cell staining and imaging

Cell staining was performed using a Biomek FX robot with a 384-well tip head. Cells were fixed and permeabilized with 5% paraformaldehyde (Sigma) and 0.2% Triton X-100 (Sigma) for ~60 min at room temperature. Nuclei and actin were stained with 2 μg/ml Hoechst 33342 (Invitrogen) and 75 ng/ml phalloidin labeled with tetramethylrhodamine isothiocyanate (Sigma) for ~60 min at room temperature. Cells were washed four times with PBS (Invitrogen), and 0.05% sodium azide (Sigma) was added for storage. Plates were sealed with aluminum seals (Corning) and stored until imaged at 4°C while protected from light. Fluorescence images were acquired with an InCell Analyzer 2000 (GE Healthcare) at 10× magnification. Each well was fully covered by four images in each of the two color channels, resulting in ~295,000 images.

## Image processing and feature extraction

Images were obtained as 16-bit TIFF images with a size of 2,048 pixels × 2,048 pixels. We adapted intensity correction, image segmentation, and feature extraction methods from previous studies, based on the R package EBImage (Pau *et al*, 2010). To remove biases due to lower illumination intensity at the image border, 150 pixels were cropped on each side. Nuclei were segmented by adaptive thresholding of the Hoechst channel images with a window size of 10 by 10 pixels. The number of segmented nuclei was used as a proxy for cell count. Using the segmented nuclei as seeds, a cell segmentation mask was generated by extending the nuclei segmentation into a threshold mask of the actin channel using a Voronoi-based propagation algorithm. Parameter and method settings are documented in the PGPC vignette. Briefly, the detected nuclei were used as seeds and expanded into masks of the cytoplasm for each cell. Morphological and texture features were extracted from the images using the segmentation masks. In total, we extracted, for each well, 385 quantitative phenotypic features (Table EV3). The data were transformed using a generalized logarithm transformation (Huber *et al*, 2002).

## Selection of non-redundant features

To select informative, non-redundant features, a stepwise dimensionality reduction and selection algorithm was employed (Laufer *et al*, 2013). Starting with cell number as an initial feature, this iterative approach fits each feature by a linear model using the selected features as predictors. The correlations between the model residuals of each replicate were used as a surrogate for the novel information that the feature contains. The feature with the highest correlation of the model residuals is selected next. This process continues until the percentage of positive model residual correlations over all features is smaller than 50%.

The final set of 20 phenotypic features was grouped into five categories. The category "DNA texture/intensity" includes intensity- and texture-related features computed from the Hoechst staining image, such as Haralick texture features. The "nuclear shape" group includes size- and shape-related features computed from the Hoechst channel, including eccentricity and nuclear radius. The "cell shape" group includes size- and shape-related features and the "actin texture/intensity" group includes intensity- and texture-related features extracted from the actin channel. The 20 phenotypic features were visualized by radar charts, which we termed phenoprints. Here, the radial distance is proportional to the variable shown. Using cell number as an example, higher distance from the origin corresponds to higher cell number.

## Quantification of chemical–genetic interactions

The data of each feature were modeled using a multiplicative model as previously described (Laufer *et al*, 2013) and robust L1 regression to estimate the effects of the cell line and compound treatment using the medpolish function of the statistics package R (http://www.r-project.org). In this iterative approach row and column medians are subtracted alternately until the change in S, the sum of absolute residuals, divided by S, falls below the defined threshold of 0.0001. The final row and column values describe the compound and cell line effect, respectively. The residuals, either having a positive or a negative value, represent the interaction coefficients. This process was performed for each replicate and each feature individually. To account for the different proliferation rates of isogenic cell lines, the cell number values on the generalized logarithmic scale were normalized using the range defined by the median of the negative control values (1) and values of the compound taxol (0) for each cell line. Values below 0 and above 1 are possible. To detect significant interactions, the values of replicates were used to perform a moderated *t*-test against the null hypothesis $\mu = 0$ using the implementation of the lmFit and eBayes functions of the limma R package (Smyth, 2004) on the interaction matrix of each feature. *P*-values were adjusted for multiple testing by controlling for the false discovery rate (FDR) using the method of Benjamini and Hochberg (1995) as previously described for the quantification of gene–gene interactions (Laufer *et al*, 2013). Significant interactions were selected by using a cutoff of 0.01 (FDR) on the adjusted *P*-values.

To predict compound mode of action, we performed hierarchical clustering with the complete linkage rule (Fig 5A). As measure of dissimilarity, we used $1 - \text{cor}(x, y)$, where x and y are the interaction profiles for two compounds and cor is the Pearson correlation coefficient.

## Phenotypic chemical–genetic interaction map

We used Cytoscape version 2.8. (Shannon *et al*, 2003) to plot the phenotypic chemical–genetic interaction map of a filtered dataset. Briefly, we removed controls and considered only those compounds that had interactions with a maximum of three out of twelve genetic backgrounds tested. We further only considered compounds that affected more than one phenotypic feature. Due to these filtering steps, five of the twelve genetic backgrounds tested (both parental HCT116 lines, *BAX*, *AKT1*, and *MEK2* KO cells) are not included in the map. A data file to produce the Cytoscape map is included in the R/Bioconductor package PGPC.

## Resolution index (ΔAUC)

To quantify the gain of information using the high-content phenotypic chemo-genomic approach over a high-content phenotypic (single cell line) or pharmacogenetic (just cell number in all cell lines) approach, we computed the resolution index ΔAUC as follows. First, the correlation of compound profiles was calculated using all features and all cell lines (genotypes and multiparametric phenotypes), all 20 selected phenotypic features of the parental HCT116 cell line P1 (multiparametric phenotypes), and just the cell number feature of all cell lines (genotypes). Second, the annotated target selectivity (Table EV1) was used to classify compound pairs into the "shared selectivity" or "no shared selectivity" class depending on whether compounds share the same target based on annotation. Chemical similarity was used to classify compound pairs into the "similar structure" or "different structure" class. For this, we used the distance matrix calculated from the compound sdf files using the ChemmineR package (Cao *et al*, 2008). Compounds were classified as "similar structure" if their structural distance as defined by the Tanimoto distance calculated by ChemmineR was below 0.6. For both approaches, the empirical cumulative distribution function (ECDF) of the compound profile correlations between compound pairs was calculated separately for each of the two classes. The

resolution index is defined as the difference of the area under the curve (ΔAUC) between the two classes for each approach.

## Analysis of drug combination data

We measured the impact on cell viability of pairwise compound combinations using fixed-dose ratio concentration kinetics and employed the CellTiterGlo assay (Promega) to determine cell proliferation and viability independently from cell number. Compounds were combined at a concentration of 20 mM each in a pairwise fashion and diluted in a 1:2 series to cover 10 concentrations. MK2206 was obtained from Santa Cruz Biotechnology. AKTi VIII was obtained from VWR International. Bendamustine, disulfiram, U0126, and PD98,059 were obtained from Sigma. Compounds were spotted in 384-well plates (Greiner) and were then diluted in RPMI using a Biomek FX robot. Briefly, HCT116 and DLD-1 cells were seeded at a concentration of 1,000 cells in 45 µl McCoy's medium into each well of 384-well plate (Greiner) and incubated for 1 day at 37°C. 5 µl of compounds was then added to cells as described before to cover a concentration range of 10–0.0195 µM. Following compound administration, cells were incubated for 3 days at 37°C and cell viability was measured via the CellTiterGlo assay (Promega) using a Mithras LB940 plate reader (Berthold Technologies). Data were analyzed using cellHTS2 (Pelz *et al*, 2010).

As compound effects E, we used $1 - \text{NPI}$, which is the normalized percentage inhibition obtained from the CellTiterGlo assay data by subtracting the value of each measurement from the average of the intensities on the plate positive controls and dividing the result by the difference between the means of the measurements on the positive and the negative controls on the plate. The raw plate reader values were logarithm-transformed before these calculations.

We quantified the unexpectedness of the effect of a compound pair by a non-interacting model (Bliss independence, BI) as previously used for large-scale compound synergism screens (Tan *et al*, 2012). In the BI model, the combined effect of the compound combination A and B is given by $E_{AB} = E_A + E_B - E_{A:B}$, where $E_A$ and $E_B$ are the single compound effects at the same dose as in the combination and $E_{A:B}$ is the interaction term, which is zero for non-interacting compounds. For each concentration, we used at least 10 measurements of $E_{AB}$ and 20 measurements each of $E_A$ and $E_B$. We estimated the interaction effect $E_{A:B}$ by inserting the means of the measurements and solving the equation for $E_{A:B}$. To test it against the null hypothesis $E_{A:B} = 0$, we employed Student's *t*-test.

## Cell-based proteasome activity assay

Compounds to be tested were spotted into 384-well plates (Greiner) at a concentration of 50 µM. ZPCK, disulfiram, CAPE, tyrphostin AG555, AG1478, and DAPH were obtained from Sigma. Bortezomib was obtained from NEB and MG132 was obtained from Merck Bioscience. HCT116 cells were seeded at a concentration of 3,000 cells in 45 µl McCoy's medium into each well of 384-well plate (Greiner) and incubated for 1 day at 37°C. Following compound administration at a final concentration of 5 µM and 0.1% DMSO, cells were incubated for 24 h at 37°C and the chymotrypsin-like, trypsin-like, and caspase-like proteasome activities were measured using the Proteasome-Glo™ Cell-Based Assay Kit according to the manufacturer's instructions (Promega) using a Mithras LB940 plate

reader (Berthold Technologies). To account for compound effects on cell proliferation, cell viability was measured via the CellTiterGlo assay (Promega). The data are normalized to the viability control CTG-assay wells on each plate (CTG was set to 1 on each plate). Based upon values corrected for cell viability, we calculated proteasome activity compared with the DMSO controls of the corresponding wells on each plate. The proteasome activity for DMSO was set to 1 for each assay. The inhibition was calculated relative to this value. Proteasome inhibition is then defined by $100*(1 - (PT/PC)/(VT/VC))$, where PT is the respective proteasome activity for each drug treatment, PC is the respective proteasome activity for control (DMSO) treatment, VT is the respective cell viability for each drug treatment, and CV is the cell viability for control (DMSO) treatment. Consequently, DMSO control determines 0% normalized proteasome inhibition. We performed a *t*-test comparing values for the compounds against the null hypothesis of zero effect.

## Western blotting

HCT116 cells were seeded at a concentration of one million cells in 2 ml McCoy's medium in each well of a 6-well plate. The next day, compounds were added in 2 ml fresh medium at a final concentration of 5 µM and 0.1% DMSO and cells were incubated for 24 h. ZPCK, disulfiram, CAPE, tyrphostin AG555, AG1478, and DAPH were obtained from Sigma. Bortezomib was obtained from NEB and MG132 was obtained from Merck Bioscience. Cells were harvested in lysis buffer and prepared for Western blotting as previously described (Kranz & Boutros, 2014). Protein concentration was measured using the BCA protein assay kit (Pierce, Thermo Scientific). Twenty microgram samples were supplemented with 5× Laemmli buffer and heated for 5 min at 96°C. Cell lysates were separated on 4–12% NuPAGE Bis/TRIS gels (Life Technologies) and transferred to Immobilon PVDF membranes (Millipore, Merck Biosciences). Antibodies used were anti-ubiquitin (clone P4D1, Cell Signaling; 1:1,000), anti-β-actin (Abcam; 1:20,000), and HRP-conjugated anti-mouse IgG2b (Southern Biotechnology; 1:10,000).

## Data availability

Complementary views on the data are available through the following avenues. The image data files are available from the BioStudies database at the European Bioinformatics Institute (EMBL-EBI) under the accession S-BSMS-PGPC1 (http://wwwdev.ebi.ac.uk/biostudies/studies/S-BSMS-PGPC1). An interactive front-end for exploration of the images is provided by the IDR database (http://dx.doi.org/10.17867/10000101). On www.bioconductor.org, the package PGPC provides an executable document with the code that was used for the analysis reported in the paper, as well as intermediate data types, such as the numeric features (https://bioconductor.org/packages/devel/data/experiment/html/PGPC.html, see Code EV1). The authors are hosting an interactive webpage to browse images and interaction profiles at http://dedomena.embl.de/PGPC.

**Expanded View** for this article is available online.

## Acknowledgements

We thank T. Mirsch and B. Schmitt for help with compound library management and robotics. We thank T. Sandmann, T. Horn, M. Billmann, B. Fischer,

and other members of the Boutros and Huber groups for helpful discussions. F.A.K. and W.H. acknowledge funding from the EC FP7 project "Systems Microscopy". Research in the Laboratory of M.B. is supported by an ERC Advanced Grant.

## Author contributions

MBr, WH, and MBo conceived and designed experiments. MBr performed experiments. FAK performed bioinformatic analysis and developed the R/Bioconductor package PGPC and the PGPCviewer with help from MBr. MBr, FAK, WH, and MBo wrote the manuscript.

## Conflict of interest

The authors declare that they have no conflict of interest.

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
