## [Review Process File · Molecular Systems Biology]

A chemical-genetic interaction map of small molecules using high-throughput imaging in cancer cells

Marco Breinig, Felix A. Klein, Wolfgang Huber and Michael Boutros

Corresponding author: Michael Boutros, German Cancer Research Center (DKFZ) and Wolfgang Huber, European Molecular Biology Laboratory (EMBL) Heidelberg

Review timeline:

Submission date:	05 July 2015
Editorial Decision:	26 August 2015
Revision received:	05 October 2015
Editorial Decision:	16 October 2015
Revision received:	30 October 2015
Accepted:	06 November 2015

Editor: Thomas Lemberger

Transaction Report:

1st Editorial Decision

26 August 2015

Thank you again for submitting your work to Molecular Systems Biology. We have now heard back from two of the three referees who agreed to evaluate your manuscript. Rather than delaying the process further, I prefer to make a recommendation now with the two available reports. As you will see from the reports below, the referees find the topic of your study of potential interest. They raise, however, several concerns, which should be convincingly addressed in a revision of the manuscript. The recommendations provided by the reviewers are very clear and refer to the need to tone down some of the claims and to clarify several aspects of the study.

If you feel you can satisfactorily deal with these points and those listed by the referees, you may wish to submit a revised version of your manuscript. Please attach a covering letter giving details of the way in which you have handled each of the points raised by the referees. A revised manuscript will be once again subject to review and you probably understand that we can give you no guarantee at this stage that the eventual outcome will be favorable.

Reviewer #1:

The manuscript "A chemical-genetic interaction map of small molecules using high-throughput imaging in cancer cells" by Breinig et al describes a new dataset with high-content phenotyping of a compound library across a panel of cell lines which includes some isogenic pairs with specific

known differences. The resulting phenotypic fingerprints appear useful in generating new hypotheses. The dataset and its analysis will undoubtedly be of high interest to readers. There are some issues in the rigor of the interpretation that are fixable. The use of Bliss independence for drug synergy measurements is a problem. There are many other addressable minor issues.

Major concerns

-The paper is unusually rich in the 20/20 hindsight that so frequently accompanies the analysis of a new large-scale dataset. This is expected to some degree, and actually quite valuable in supporting the idea that a dataset will be useful for hypothesis generation. Here the reader's credulity is stretched when it is suggested that remarkably specific conclusions about drug mechanism can be derived solely from this data, when in fact what is being noted is that the data is simply consistent with already-known mechanisms. Obviously the dataset will be useful, but the implications about what could have been learned from this dataset alone need to be toned down enormously. A (non-exhaustive) list of examples follows:

+"Of note, our resource further informed about different bioactivities for structurally highly related compounds (arrowheads, off-diagonal). 5-azacytidine and ara-c did not directly cluster together (arrowhead 1). Mechanistically, this finding is explained by the fact that, while both compounds are cytosine analogues, only 5-azacytidine inhibits DNA methyltransferases (Christman, 2002). [this is a post-hoc rationalization, not something any user could have been "informed about" from the dataset alone]

+"the anti-helmitic compound niclosamide, which uncouples oxidative phosphorylation (MacDonald et al., 2006; Weinbach and Garbus, 1969), clustered together with rottlerin, a compound thought to target PKC. Our data indicate that rottlerin's mode-of-action is in fact linked to uncoupling oxidative phosphorylation, which is in accordance with recent findings..." [post-hoc rationalization]

+"Of note, our map revealed connected biological processes perturbed by compounds that target different effector molecules (C11-16). For instance, we observed a cluster including inhibitors of folate metabolism and the DNA methyltransferase inhibitor 5-azacytidine (C16). Additionally, we found the iron chelator phenanthroline within this cluster. Associations between iron and folate metabolism have been reported (Oppenheim et al., 2000) and folate metabolism is linked to DNA methylation, which is an important regulator of gene transcription (Crider et al., 2012). [this seems a tenuous thread to rationalize co-clustering. Many distinct biological processes are 'linked' to a similar extent but these linkages do not result in similar phenotypes. This is fine as a clearly-labeled post-hoc rationalization, but no suggestion should be made that this dataset in and of itself could "reveal" these connections]

In these and other examples, it would also lend more support to discuss what phenotypes/genes/cell lines are shared where there is co-clustering or where there are differences in cases of lack of co-clustering.

-Drug-drug synergy is measured by Bliss independence, which is a terrible synergy measure by which a drug can be found to synergize with itself. Imagine two half-concentration doses of the same drug, where each half-concentration dose is just to the low-response end from the inflection point of a sigmoidal dose-response curve. Even if though these are (by definition) additive drugs, doubling the dose will more than double the response given the sigmoidal shape. Most drugs have non-linear dose-response curves so the possibility of this scenario is the rule rather than the exception.

Minor concerns

-Fig 1 D shows human-readable labels describing mathematically-derived combination phenotypes, but where did these labels come from? How does the reader know whether a cell number measure that is farther from the origin is an increase vs a decrease in cell number. Reference is made to phenoprints that show apoptotic behaviour. How should a non-cell-biologist reader (or cell biologist, for that matter), glean this from the phenoprint?

-Might contrast with the "Rosetta compendium" or "Connectivity Map" expression studies as prior examples of useful high-content phenotyping/fingerprinting used for understanding drug mechanisms

- Should make clear which cell line "knockout" mutations are heterozygous and which are homozygous
- PD98.059 and DMSO control are said to be "not completely similar". In what respect is there a significant (and therefore reportable) difference?
- MEK1 KO having more interactions than MEK2 KO cells need not imply distinct functions. This might also be seen for two paralogs with identical function where one of them is more highly expressed
- "we observed interactions between the PI3K inhibitor wortmannin and KRAS wt cells" What does it mean for a drug to interact with just one allele of a gene? I thought that interaction was always between a drug and an allele, with wt being the baseline used for making that judgment. There are other examples of this kind of statement.
- No details are given on the "unsupervised clustering" method. It is apparently hierarchical, but what linkage method, what distance measure?
- Some of the knockout mutations presumably have a profound effect on cell line growth rate. It would be of great interest to do analysis of variation and report how much of the variation in phenotypes between cell lines can be explained by cell growth rate, then how much of the residual variation by the known genotypic differences between pairs of cell lines, and how much of that residual variation is attributable to the unknown genotypic differences between cell lines.
- The authors acknowledge that growth rate can cause major differences not just in phenotype but in gene-drug interactions. It would be good to know how many gene-drug interactions remain when gene-drug interactions are calculated not with phenotypes but instead by the residual phenotype after doing growth-based prediction of phenotypes.
- The fact that some predictions of synergy were confirmed is nice, but a true demonstration that the dataset predicts synergy requires more tests of non-predicted drug-drug pairs (synergy is quite common, especially by Bliss independence!). A related earlier study in yeast (Cokol et al MSB 2011) found that while genetic interactions among drug target genes predicted synergy, these predictions didn't perform any better than chance once the baseline drug-interaction rate of each drug was taken into account. I'm not suggesting that dozens to hundreds of new drug interactions need to be tested, just that this caveat needs to be given and the claims toned down.
- Empty phrases like "of note" should be removed
- No parameter settings are provided for threshold masking, segmentation masking, or variance-stabilizing transformations
- In the selection of non-redundant features, how was the initial starting feature chosen?
- "In this iterative approach row and column median values are subtracted alternately until the proportional change of the absolute residuals falls below a defined threshold." [Not clear how this proportional change is defined]
- What were the parameters for the regularized t-test
- "We provide a re-usable computational analysis workflow" [This is not described sufficiently well to make this a selling point of the paper. Maybe take this comment out and save it for an applications note?]
- The results of Benjamini-Hochberg analysis should be referred to as FDR estimates or q-values rather than adjusted p-values. The authors are not alone in this practice, but p-values should be reserved for the (less useful) practice of controlling Type I error rate.
- What is "overplotting"?

- "cytoscape"

-Pg 20. "structural distance" between compounds is not defined

-Not clear how normalization of proteasome assays by cell viability was performed.

Reviewer #2:

Breinig et al. report here the development of a chemical-genetic interaction map based on imaging features extracted through high-content screening in a set of isogenic HCT116 cells. Such a phenotypic profiling resource can be very valuable to examine compound mechanism of action, similarity to other compounds, and filtering out compounds with potential deleterious side effects. This manuscript is well-written and presented, but I feel that there are a number of points of clarification that should be made before the manuscript is suitable for publication.

1. There have been other very similar efforts in this precise area, using more cellular stains (e.g. Gustafsdottir et al., PLOS ONE 2013). It is imperative that the authors discuss the differences and/or advantages of this method.
2. I could not find a list of the total 395 features extracted from each well, which would be helpful.
3. 80% of the features had correlation >0.7 between duplicates, which did not seem exceptionally high. Was there something about the other 85 features that contributed to the lack of correlation?
4. Cells were treated for 2 days with 5 μ M of each compound, which seems long and high for many of these common bioactive compounds. I would imagine many toxic effects. Perhaps a shorter treatment would increase correlation? It might be worth discussing in the manuscript. For example, it is difficult to interpret the results of wortmannin used at 5 μ M, as it is hitting quite a number of targets at that concentration.
5. How are the phenoprints compared quantitatively? The impression from the paper is that they are visually similar by eye, but that cannot be sustainable across so many compound treatments.
6. It is mentioned in the methods that serial dilutions of each compound were made, but there is not discussion of the effects of dose on phenotypes. Were the compounds tested in dose? If not, that point needs to be clarified.
7. Images need to be larger, or include some insets of higher magnification. It is impossible to see what the authors are trying to convey.
8. Similarly, in Figure S2, it is not clear what the reader is supposed to see in this image. A control should be included for greater clarity.
9. The lower number of chemical-genetic interactions based on cell number is confusing, as the other publications looking at chemical-genetic interactions using cell viability see quite a few interactions. Some discussion of the lower number here would be helpful.
10. In the proteasome inhibition assay, what is the effect of DMSO alone? Is it pegged at zero? This would help interpret the graph shown in Figure 6.

We thank the reviewers for their effort spent on our paper and for their helpful comments. In the following we provide a point-by-point reply. We have revised the text and the figures accordingly. Changes in the text are highlighted in yellow.

Reviewer #1:

The manuscript "A chemical-genetic interaction map of small molecules using high-throughput imaging in cancer cells" by Breinig et al. describes a new dataset with high-content phenotyping of a compound library across a panel of cell lines which includes some isogenic pairs with specific known differences. The resulting phenotypic fingerprints appear useful in generating new hypotheses. The dataset and its analysis will undoubtedly be of high interest to readers. There are some issues in the rigor of the interpretation that are fixable. The use of Bliss independence for drug synergy measurements is a problem. There are many other addressable minor issues.

We thank the reviewer for this positive feedback.

Major concerns

-The paper is unusually rich in the 20/20 hindsight that so frequently accompanies the analysis of a new large-scale dataset. This is expected to some degree, and actually quite valuable in supporting the idea that a dataset will be useful for hypothesis generation. Here the reader's credulity is stretched when it is suggested that remarkably specific conclusions about drug mechanism can be derived solely from this data, when in fact what is being noted is that the data is simply consistent with already-known mechanisms. Obviously the dataset will be useful, but the implications about what could have been learned from this dataset alone need to be toned down enormously.

We fully agree regarding the logical difference between post-hoc rationalization and discovery of something new and unexpected. We thank the reviewer for pointing out the imprecise language of the previous manuscript. We have carefully gone over the whole manuscript and revised the language accordingly.

A (non-exhaustive) list of examples follows:

+"Of note, our resource further informed about different bioactivities for structurally highly related compounds (arrowheads, off-diagonal). 5-azacytidine and ara-c did not directly cluster together (arrowhead 1). Mechanistically, this finding is explained by the fact that, while both compounds are cytosine analogues, only 5-azacytidine inhibits DNA methyltransferases (Christman, 2002). [this is a post-hoc rationalization, not something any user could have been "informed about" from the dataset alone]

The wording is now: *"There were also instances of structurally highly related compounds with different bioactivities (arrows, off-diagonal). For example, the two cytosine analogues 5-azacytidine and ara-c did not directly cluster together (arrow 1), consistent with the fact that only 5-azacytidine inhibits DNA methyltransferases (Christman, 2002)."* (Page 10).

+ "the anti-helmitic compound niclosamide, which uncouples oxidative phosphorylation (MacDonald et al., 2006; Weinbach and Garbus, 1969), clustered together with rottlerin, a compound thought to target PKC. Our data indicate that rottlerin's mode-of-action is in fact linked to uncoupling oxidative phosphorylation, which is in accordance with recent findings..." [post-hoc rationalization]

We have revised the language. The wording is now: *"The map further allowed us to infer primary mode-of-action and off-target effects of compounds (C17, C18 see below). For example, C17 clustered the anti-helmitic compound niclosamide, which uncouples oxidative phosphorylation (MacDonald et al., 2006; Weinbach and Garbus, 1969), and rottlerin, a compound thought to target PKC. This result is in accordance with the finding that rottlerin uncouples oxidative phosphorylation and supports the view that the classification of rottlerin as a PKC δ inhibitor is incorrect (Soltoff, 2001)."* (Page 11).

+ "Of note, our map revealed connected biological processes perturbed by compounds that target different effector molecules (C11-16). For instance, we observed a cluster including inhibitors of folate metabolism and the DNA methyltransferase inhibitor 5-azacytidine (C16). Additionally, we found the iron chelator phenanthroline within this cluster. Associations between iron and folate metabolism have been reported (Oppenheim et al., 2000) and folate metabolism is linked to DNA methylation, which is an important regulator of gene transcription (Crider et al., 2012).

[this seems a tenuous thread to rationalize co-clustering. Many distinct biological processes are 'linked' to a similar extent but these linkages do not result in similar phenotypes. This is fine as a clearly-labeled post-hoc rationalization, but no suggestion should be made that this dataset in and of itself could "reveal" these connections]

We have revised the language. The wording is now: *"The map further contained several instances of clusters of compounds that target different effector molecules in connected biological processes (C11-16). For example, C16 included inhibitors of folate metabolism, the DNA methyltransferase inhibitor 5-azacytidine and the iron chelator phenanthroline. Associations between iron and folate metabolism have been reported (Oppenheim et al., 2000), and folate metabolism is linked to DNA methylation (Crider et al., 2012)."* (Page 11).

To further address the reviewer's concern, we have additionally revised the language on following pages:

Page 4. The wording is now: *"Exploring the PGPC, we could see instances of pathway crosstalk..."*

Page 9. The wording is now: *"In support of this interpretation, a link between..."*

Page 9. The wording is now: *"...to derive insights how drugs perturb genetic networks..."*

Page 10. The wording is now: *"...clusters could in some cases be explained by..."*

Page 16. The wording is now: *"...we predicted associations between drugs..."*

In these and other examples, it would also lend more support to discuss what phenotypes/genes/cell lines are shared where there is co-clustering or where there are differences in cases of lack of co-clustering.

Thank you for this suggestion. We have extended the text to include several examples that respond to this question and we added a new supplementary figure (Supplementary Fig 12).

The revised text is: *"While in some instances the correlation of drug profiles was a result of coordinated subtle covariation across multiple cell lines and phenotypes, in other cases it appeared to be driven by the similarity of distinctive phenotypes of individual isogenic cell lines. For example, and as already introduced above, the CK2 inhibitors DMAT and TBBz (Fig 5A, C5) affected nuclear shape features specifically in cells with only the wildtype copy of PI3KCA (HCT116^{PI3KCA wt +/- mt -}) (Supplementary Fig 12 and Fig 3), whereas the compounds ARP101 and YC-1 in C15 affected distinct phenotypic features in cells with only the wildtype copies of CTNNB1 (HCT116^{CTNNB1 wt +/- mt -}) and PI3KCA (HCT116^{PI3KCA wt +/- mt -}) (Supplementary Fig 12). Further, compounds in C18 had distinctive interactions with MEK1 KO cells (HCT116^{MAP2K1 -/-}) (Supplementary Fig 12 and Fig. 3)." (Page 11).*

We further included: *"...the compounds we observed to impair proteasome function (Fig 5A C18 and Fig 6A) primarily interacted with MEK1 KO cells (Fig. 3), which might indicate that the combination of MEK inhibitors with compounds that impair proteasome activity could generally be effective." (Page 15).*

We have also added: *"There were instances in which drug-gene interactions using only a viability phenotype, i.e. cell number, grouped together related compounds, including microtubule inhibitors such as vinblastine and nocodazole (Supplementary Fig S13, genotypes). Likewise, there were instances in which phenotypic profiling of drugs using solely the parental HCT116 cell line grouped together related compounds, including the MEK inhibitors PD98059 and U0126 (Supplementary Fig S13, multiparametric phenotypes). However, associations between other compounds, including the structurally highly related glucocorticoids betamethasone and beclomethasone (Fig 5A, C4) were only revealed by the combined approach (Supplementary Fig S13, genotypes and multiparametric phenotypes)." (Page 12).*

-Drug-drug synergy is measured by Bliss independence, which is a terrible synergy measure by which a drug can be found to synergize with itself. Imagine two half-concentration doses of the same drug, where each half-concentration dose is just to the low-response end from the inflection point of a sigmoidal dose-response curve. Even if though these are (by definition) additive drugs, doubling the dose will more than double the response given the sigmoidal shape. Most drugs have non-linear dose-response curves so the possibility of this scenario is the rule rather than the exception.

We agree with the reviewer and are equally concerned about this property of the Bliss independence (BI) model, which is indeed also shared by the highest single agent (HSA) model. Detecting and quantifying drug-drug interactions is a subtle problem in general, and we are aware of some of the extensive literature. Here, we do not aim to solve this problem

in generality, but only for two exemplary cases, in which a more qualitative approach is possible:

- In the case of bendamustine and AKT inhibitors (AKTi VIII and MK-2206), the interaction data at 10 μM concentration are shown in Fig 4B. We draw confidence on the significance of this interaction by comparison with the data for the MEK inhibitor, shown in Fig S10A,B. Moreover, analogous results were observed in two different cell lines, HCT116 (shown in Figs 4B and S10A,B), and DLD1 (Fig S10C-F). For instance, in Fig S10D, MK-2206 alone even had a slightly pro-proliferative effect on the cells, while the same dose together with bendamustine more than doubles the effect on cell growth of bendamustine alone. See also Review Figure 1.
- In the case of disulfiram and MEK inhibition (U0126 and PD98.059), Fig 4D shows that both of the MEK inhibitors have no (or if any, a pro-proliferative) effect on the cells, while together with disulfiram the effect is about twice as high as for disulfiram alone.

In other words, we acknowledge that drug synergy is difficult to detect and quantify when both drugs individually already have comparably strong effects. However, in the cases discussed in our manuscript, one of the two drugs has no (or negligible) effect alone, but strongly amplifies the effect of the other drug when combined.

Review Figure 1: The combination of bendamustine with AKT inhibitors is effective, whereas the combination of bendamustine with MEK inhibitors is not. Error bars show mean \pm s.e.m, where $n \geq 3$ with at least three independent experiments that determined cell viability using the CellTiterGlo assay in HCT116 cells. BI significance is shown, asterisk (*) indicates $p < 0.05$.

We have modified the text to better clarify the effect of single drug treatments to address the reviewer's concern.

Page 2: "...we demonstrate that MEK inhibitors amplify the viability effect of the clinically used anti-alcoholism drug disulfiram."

Page 9: *“We therefore tested the combination of bendamustine with small molecule AKT inhibitors and observed that while AKT inhibition alone had a negligible effect on viability of the parental HCT116 cells, in combination with bendamustine the effect was significantly stronger than the bendamustine single agent treatment.”*

Page 10: *„The combination of disulfiram and MEK inhibitors resulted in significantly stronger viability reduction compared to disulfiram alone, while MEK inhibitors alone had no detectable effect...“.*

Minor concerns

-Fig 1 D shows human-readable labels describing mathematically-derived combination phenotypes, but where did these labels come from?

How does the reader know whether a cell number measure that is farther from the origin is an increase vs a decrease in cell number. Reference is made to phenoprints that show apoptotic behaviour. How should a non-cell-biologist reader (or cell biologist, for that matter), glean this from the phenoprint?

An explanation of the 20 numbers visualized in each of the radar charts of Fig 1 is given in Fig S3, and their interpretation by the human readable terms is based on the nature and input data of the mathematical formulae implementing them. Conventionally, in radar charts the radial distance is proportional to the variable shown, so higher distance from the origin represents higher cell number. Readers might also infer this convention by inspecting the given example images together with the radar charts.

We have revised the text to clarify this question: *“The final set of 20 phenotypic features was grouped into 5 categories. The category ‘DNA texture/intensity’ includes intensity- and texture-related features computed from the Hoechst staining image, such as Haralick texture features. The ‘nuclear shape’ group includes size- and shape-related features computed from the Hoechst channel, including eccentricity and nuclear radius. The ‘cell shape’ group includes size- and shape-related features and the ‘actin texture/intensity’ group includes intensity- and texture-related features extracted from the actin channel. The 20 phenotypic features were visualized by radar charts, which we termed phenoprints. Here, the radial distance is proportional to the variable shown. Using cell number as an example, higher distance from the origin corresponds to higher cell number.”* (Page 20).

Our intention for the phenoprints is not to replace the images, but to serve as a compact visualization of the 20 features that we use to measure image similarity. In our experience, we have found them useful to get a rapid overview over trends and changes in our data without always needing to inspect a large number of microscopy images.

We have now adjusted the text to avoid the (mis)understanding that we ask human readers to associate specific forms of phenoprints with specific processes such as apoptosis. The wording is now: *“...We visualized them by radar charts, which we term phenoprints (Fig 1D). We explored these charts together with the original images to assess their ability to report specific drug-induced morphological changes. Treatment of the parental HCT116 cells with microtubule-targeting compounds caused apoptosis (Fig 1F and*

G), and topoisomerase inhibitor treatment resulted in increased nuclear and cellular size as compared to DMSO treated cells (compare Fig 1 E with Fig 1H and I), a phenotype attributed to mitotic catastrophe (Maskey et al., 2013). Further examples of characteristic phenotypes included cell death with aberrantly shaped nuclei and locally enhanced actin intensity in the few remaining cells (Ouabain, Fig 1J) as well as elongated cells (Rottlerin, Fig 1K). Overall, phenoprints served as compact visualizations of drug-induced phenotypes. Drugs sharing the same target resulted in similar phenotypes and had similar phenoprints, as demonstrated by microtubule-targeting compounds (Fig 1F and G), and topoisomerase inhibitors (Fig 1H and I).“ (Page 6).

-Might contrast with the "Rosetta compendium" or "Connectivity Map" expression studies as prior examples of useful high-content phenotyping/fingerprinting used for understanding drug mechanisms

We added a comment into the Discussion section: “...integration of emerging pharmacogenetic and phenotypic resources (Barretina et al., 2012; Basu et al., 2013; Garnett et al., 2012; Kleinstreuer et al., 2014; Young et al., 2008) and complementary strategies such as transcription profiling as a means to infer drug mode-of-action (Lamb et al., 2006) will accelerate the development of efficient genotype-stratified therapeutics and a better understanding of side-effects...” (Page 17).

- Should make clear which cell line "knockout" mutations are heterozygous and which are homozygous

In the previous version of our manuscript, we provided an overview of the isogenic cell lines employed in our study in Supplementary Figure S1, showing their respective genotypes. To better clarify the genetic background of the isogenic cell lines, we have now extended the cell line names and provide this information in the Material and Methods section as well as the Results section and Figure legends where appropriate: Pages 6, 8, 9, 10, 18 and Fig 2, 3, 4.

Previous cell line name	Extended cell line name
parental HCT116	HCT116 ^{CTNNB1 wt +/- mt + ; KRAS wt +/- mt + ; PI3KCA wt +/- mt +}
CTNNB1 wt	HCT116 ^{CTNNB1 wt +/- mt -}
KRAS wt	HCT116 ^{KRAS wt +/- mt -}
PI3KCA wt	HCT116 ^{PI3KCA wt +/- mt -}
PTEN KO	HCT116 ^{PTEN -/-}
AKT1 KO	HCT116 ^{AKT1 -/-}
AKT1/2 KO	HCT116 ^{AKT1 -/- ; AKT2 -/-}
MEK1 KO	HCT116 ^{MAP2K1 -/-}
MEK2 KO	HCT116 ^{MAP2K2 -/-}
P53 KO	HCT116 ^{TP53 -/-}
BAX KO	HCT116 ^{BAX -/-}

For example, the isogenic cell line previously referred to as KRAS wt represents the HCT116 cell line in which the mutant KRAS allele has been targeted leaving only the respective KRAS wt allele and as such is now also referred to as HCT116^{KRAS wt +/- mt -}.

- PD98.059 and DMSO control are said to be "not completely similar". In what respect is there a significant (and therefore reportable) difference?

We agree with the reviewer that the presentation of the image and phenoprint for PD98.059 was unclear; indeed it is also not necessary for the message that we want to convey, and in the interest of brevity and clarity we have removed it.

- MEK1 KO having more interactions than MEK2 KO cells need not imply distinct functions. This might also be seen for two paralogs with identical function where one of them is more highly expressed

We thank the reviewer for pointing out this possibility. We have changed the text: *"Possible reasons for this observation include different levels of expression of MEK1 and MEK2, and some degree of functional specialization between MEK1 and MEK2 (Catalanotti et al., 2009; Scholl et al., 2009)."* (Page 8).

- "we observed interactions between the PI3K inhibitor wortmannin and KRAS wt cells" What does it mean for a drug to interact with just one allele of a gene? I thought that interaction was always between a drug and an allele, with wt being the baseline used for making that judgment. There are other examples of this kind of statement.

We assume that the nomenclature "KRAS wt" in the previous manuscript was misleading and led to the misunderstanding raised here. The cell line that we previously referred to as "KRAS wt" is a genetically engineered line derived from the parental HCT116^{KRAS wt +/- mt +} by knockout of the mutant (mt) allele, therefore retaining only a wildtype copy of KRAS. However it is not the baseline (parental) line that serves as the reference for calling interactions. We hope that the extended nomenclature for the isogenic cell lines (as discussed above) will clarify this point.

- No details are given on the "unsupervised clustering" method. It is apparently hierarchical, but what linkage method, what distance measure?

We apologise that this information was initially only in the computational workflow (vignette). We added the information to the Materials and Methods section: *"To predict compound mode-of-action, we performed hierarchical clustering with the complete linkage rule (Fig 5A). As measure of dissimilarity we used $1 - \text{cor}(x, y)$, where x and y are the interaction profiles for two compounds and cor is the Pearson correlation coefficient."* (Page 21).

To further clarify our approach, we have additionally revised the Results part to read: *"Next, we analyzed the similarity of interaction profiles by unsupervised clustering, using one minus the correlation coefficient of drug profiles as a measure of dissimilarity."* (Page 10).

-The authors acknowledge that growth rate can cause major differences not just in phenotype but in gene-drug interactions. It would be good to know how many gene-drug

interactions remain when gene-drug interactions are calculated not with phenotypes but instead by the residual phenotype after doing growth-based prediction of phenotypes.

-Some of the knockout mutations presumably have a profound effect on cell line growth rate. It would be of great interest to do analysis of variation and report how much of the variation in phenotypes between cell lines can be explained by cell growth rate, then how much of the residual variation by the known genotypic differences between pairs of cell lines, and how much of that residual variation is attributable to the unknown genotypic differences between cell lines.

The reviewer refers to the fact that (i) some of the gene knock-out mutations in the cell line panel investigated have an effect on growth rate, and (ii) that some of the drug-gene interactions that we report could be caused via this growth rate effect (and thus be relatively indirect), while others could rely on more direct mechanisms, e.g. the drug interfering directly with the signalling pathways supported by the gene product.

The reviewer suggests that we could modify our analysis to somehow disentangle these two types of interactions (indirect via growth rate, or otherwise) by an Analysis of Variance (ANOVA) type technique. In response, we would like to make the following points:

1. We have extended the manuscript to make the issue of direct and indirect mechanisms of gene-drug interactions more apparent to the readers.
2. We are currently already performing an ANOVA analysis that regresses out the cell line effect (see Methods), and that addresses the problem to some extent.
3. ANOVA however cannot fully disentangle direct and indirect effects, and
4. a proper treatment of such a disentanglement is a research problem for which no reliable and generally applicable methods exist. We are aware of claims such as those made by B. Barzel & A.-L. Barabási, Network link prediction by global silencing of indirect correlations, *Nature Biotechnology* 31, 2013, p 720–725, and S. Feizi, D. Marbach, M. Médard & M. Kellis, Network deconvolution as a general method to distinguish direct dependencies in networks, *Nature Biotechnology* 31, 2013, p 726–733, as well as of extensive discussions on merits and limitations of these and similar approaches. We consider such method development outside of the scope of this paper.

Ad 1, the isogenic cell lines employed indeed have not only different morphologies (as shown in Supplementary Figure S4) but also different proliferation rates. To highlight the differential growth behaviour between isogenic cell lines, we have included a new plot into Supplementary Figure S7 and modified the relevant sentence to read: *“We also noted a trend towards higher number of interactions involving cell lines in which the genotype itself had a pronounced phenotypic effect, including cell number (e.g. CTNNB1 wt cells; Fig 2E and Supplementary Fig 4 and 7A and B)”* (Page 7).

Ad 2., we are indeed considering the differential growth of isogenic cell lines for our analysis of interactions by normalizing our data for the different proliferation rates (see Material and Methods). Moreover, a further containment of growth-rate mediated effects is achieved by the stepwise feature selection algorithm that we apply, which selects maximally non-collinear phenotypic features; therefore the 19 phenotypic features other than cell number are by construction already less affected by growth-rate effects.

To better clarify our approach, we have further modified the text to read: *“Briefly, the approach accounts for baseline genotype and drug effects with an ANOVA-type approach...”* (Page 7).

- The fact that some predictions of synergy were confirmed is nice, but a true demonstration that the dataset predicts synergy requires more tests of non-predicted drug-drug pairs (synergy is quite common, especially by Bliss independence!). A related earlier study in yeast (Cokol et al MSB 2011) found that while genetic interactions among drug target genes predicted synergy, these predictions didn't perform any better than chance once the baseline drug-interaction rate of each drug was taken into account. I'm not suggesting that dozens to hundreds of new drug interactions need to be tested, just that this caveat needs to be given and the claims toned down

We thank the reviewer for raising this issue and referring to the excellent Cokol et al. paper. It points out the prevalence of non-specific promiscuous synergy in the drug combinations that they studied, and hence that ‘predicting’ a drug-drug interaction may not always be terribly specific or interesting. Indeed, we also find in our dataset that certain tested compounds show interactions in a large fraction of the isogenic cell lines. However this is not the case for Bendamustine and Disulfiram, the drugs that we considered to test the prediction of effective drug combinations (see Figures 3 and 4, Supplementary Figure S6, and Review Figure 2). Hence, while the reviewers’ concern is very valid in general, we think that the specific cases described in this manuscript are still worth reporting.

We have included a reference to the paper by Cokol et al. and point out in the Discussion section: *“More research is needed for a fair assessment of prediction performance, since parameters such as prediction sensitivity and specificity need to be calibrated depending on a drug’s single-agent activity, polypharmacology and its interaction ‘promiscuity’ (Cokol et al., 2011).”* (Page 15).

We have further addressed the reviewer’s concern by toning down the generalized “prediction of synergy” claim. The wording is now:

Page 2: *“...the generation of hypotheses on drug combinations and synergism...”*

Page 2: *“...MEK inhibitors amplify the viability effect of the clinically used anti-alcoholism drug disulfiram.”*

Page 9 and 34: *“Extrapolating drug-gene interactions to drug-drug combinations.”*

Page 9: *“...predict effective drug-drug combinations.”*

Page 22: *“Analysis of drug combination data.”*

Page 23: *“We quantified the unexpectedness of the effect of a compound pair...”*

Review Figure 2: “Specifically interacting” versus “promiscuously interacting” compounds. Example compound interaction spectra are shown (see also manuscript Figure 4 and Supplementary Figure S6). Bendamustine shows multiple-feature interactions primarily in AKT1/2 KO cells and Disulfiram shows multiple-feature interactions primarily in MEK1 KO cells. In contrast, Thapsigargin, Static and β -Lapachone show multiple-feature interactions in almost all of the 12 cell lines investigated. These findings suggest that the interactions for Bendamustine and Disulfiram are relatively specific for one genetic background whereas the latter three compounds have a more promiscuous interaction pattern.

- Empty phrases like "of note" should be removed

We have removed empty phrases like “of note”, “indeed”, “in fact”.
Pages: 6, 7, 8, 9, 10, 12.

- No parameter settings are provided for threshold masking, segmentation masking, or variance-stabilizing transformations

This information is provided fully, and in a computationally reproducible (by anyone with a computer) way, in the provided R vignette. We have considered adding this information also in natural language form to the Materials and Methods section, but have found that it was

not possible to present sufficient detail without essentially recapitulating the code, yet that such text still remains unsatisfactory due to remaining ambiguities. We therefore prefer the solution with the computer code, given that the code is relatively straightforward and assuming that readers interested in that aspect will be able to parse it.

- In the selection of non-redundant features, how was the initial starting feature chosen?

Here we followed C. Laufer, B. Fischer, W. Huber and M. Boutros, Measuring genetic interactions in human cells by RNAi and imaging. Nature Protocols, 9:2341 (2014). Briefly, cell number is selected as a starting feature due to its prominence and interpretability, and to make our data as comparable as possible with other assays that record solely cell viability or growth. We have added this information to the Materials and Methods section: *“Starting with cell number as an initial feature, this iterative approach fits each feature by a linear model using the selected features as predictors.”* (Page 20).

- “In this iterative approach row and column median values are subtracted alternately until the proportional change of the absolute residuals falls below a defined threshold.” [Not clear how this proportional change is defined]

This is simply the change in the sum of absolute residuals divided by the previous sum; this quantity is used in many iterative fitting algorithms. We have rephrased: *“In this iterative approach row and column medians are subtracted alternately until the change in S , the sum of absolute residuals, divided by S , falls below the defined threshold of 0.0001.”* (Page 20).

- What were the parameters for the regularized t-test

We apologise that this information was initially only in the computational workflow (vignette). We added the information to the Materials and Methods section: *“To detect significant interactions, the values of replicates were used to perform a moderated t-test against the null hypothesis $\mu = 0$ using the implementation of the *lmFit* and *eBayes* functions of the *limma* R package (Smyth, 2004) on the interaction matrix of each feature...”* (Page 21).

- “We provide a re-usable computational analysis workflow” [This is not described sufficiently well to make this a selling point of the paper. Maybe take this comment out and save it for an applications note?]

We deleted this comment.

We nevertheless believe that the provision of an open-source software package that can be used to perform and recapitulate all analysis steps presented in the manuscript is of high importance in the context of large-data management and interpretation. We therefore state: *“To foster reproducibility, we developed an R package named PGPC containing all data and a ‘vignette’, i.e. an open source software script that performs the complete analysis presented in this study starting from the extracted feature values and reproducing all figures, tables and other quantitative results.”* (Page 19).

- The results of Benjamini-Hochberg analysis should be referred to as FDR estimates or q-values rather than adjusted p-values. The authors are not alone in this practice, but p-values should be reserved for the (less useful) practice of controlling Type I error rate.

We have now used FDR instead of adjusted p-values.

- What is "overplotting"?

We have deleted the word "overplotting".

- "cyctoscape"

Corrected.

- Pg 20. "structural distance" between compounds is not defined

We define "structural distance" between compounds as the Tanimoto distance calculated via the R/Bioconductor ChemmineR package. In the previous manuscript version, this information was given only in the computational workflow (vignette). We have now added this information into the Materials and Methods section: "...*structural distance as defined by the Tanimoto distance calculated by ChemmineR...*" (Page 22).

- Not clear how normalization of proteasome assays by cell viability was performed.

We apologise for not previously describing this more directly. This information was initially only in the computational workflow (Vignette). Briefly, we anticipated that proteasome inhibition could impair cell viability (even for the rather short incubation time of 24h that we used in these experiments), which would confound the population-based measurement of intracellular proteasome activity in our assays, if not accounted for. In other words (using the extreme as an example), a drug that reduces cell viability by 100% will consequently also reduce intracellular proteasome activities by 100%, independent of its direct effect on the proteasome.

Consequently, we investigated the effect of all tested drugs on cell viability in parallel to respective intracellular proteasome activities. In our CellTiterGlo-based cell viability assay, we saw that proteasome inhibitor treatment indeed reduced viability for Bortezomib and MG132 (see Review Figure 3). We therefore normalized the proteasome activity measurements by dividing them through the normalized cell viability measurements to account for the impact of loss-of-viability on the cell population.

Review Figure 3: Viability effect of proteasome inhibitors.

Cell viability was measured using the CellTitreGlo (CTG) assay after 24h (y-axis). Along the x-axis, symbols are sorted by treatment, which is also indicated by symbol colour. Symbol shape indicates 5 experimental replicates (referred to as 'plateName' in the plot). MG132 and Bortezomib impair HCT116 cell viability, and this needs to be accounted for when interpreting results of intracellular proteasome activity measurements.

We added the following to the Materials and Methods section: *“To account for compound effects on cell proliferation, cell viability was measured via the CellTiterGlo assay (Promega). The data is normalized to the viability control CTG-assay wells on each plate (CTG was set to 1 on each plate). Based upon values corrected for cell viability, we calculated proteasome activity compared with the DMSO controls of the corresponding wells on each plate. The proteasome activity for DMSO was set to 1 for each assay. The inhibition was calculated relative to this value.*

Proteasome inhibition is then defined by $100 \times (1 - (PT/PC) / (VT/VC))$ where PT is the respective proteasome activity for each drug treatment, PC the respective proteasome activity for control (DMSO) treatment, VT is the respective cell viability for each drug treatment, and CV is the cell viability for control (DMSO) treatment. Consequently, DMSO control determines 0% normalized proteasome inhibition. We performed a t-test comparing values for the compounds against the null hypothesis of zero effect.” (Page 23).

Reviewer #2:

Breinig et al. report here the development of a chemical-genetic interaction map based on imaging features extracted through high-content screening in a set of isogenic HCT116 cells. Such a phenotypic profiling resource can be very valuable to examine compound mechanism of action, similarity to other compounds, and filtering out compounds with potential deleterious side effects. This manuscript is well-written and presented, but I feel that there are a number of points of clarification that should be made before the manuscript is suitable for publication

We thank the reviewer for this positive feedback.

1. There have been other very similar efforts in this precise area, using more cellular stains (e.g. Gustafsdottir et al., PLOS ONE 2013). It is imperative that the authors discuss the differences and/or advantages of this method.

The approach described by Gustafsdottir et al. provides a framework to increase multiplexing capabilities for HC screens and we thank the reviewer for making us aware of this publication. Indeed, the experimental approach that we used for the present work could be extended in this way, it is likely that it would benefit, by having a greatly expanded phenotypic search space for interactions.

We have now included a statement regarding the use of additional cellular stains and why comprehensive multiplexing approaches such as those described by Gustafsdottir et al. can be advantageous: *"...Several extensions of the phenotypic pharmacogenetic screening assay presented here will be desirable. ... Moreover, the use of additional markers of cellular components for high-content imaging could further increase the phenotypic search space for interactions relevant to an even broader range of biological processes. For example, Gustafsdottir et al. developed a multiplexing protocol that allows for the detection of seven distinct cell components using six stains and imaging five channels (Gustafsdottir et al., 2013)."* (Page 16).

2. I could not find a list of the total 395 features extracted from each well, which would be helpful.

This information was initially only given in the computational workflow (vignette). We have now included a list of all features as Supplementary Table III. The list includes original feature names as obtained from our automated feature extraction using EBIImage, as well as human-readable feature names for a subset of features that we present in the manuscript, alongside information about the correlation between replicates for all features (see below).

In the process of making this addition, we noticed that we mistakenly had stated the number of features as 395 in the previous manuscript version, whereas it is actually 385. We apologise for this mistake and thank the reviewer for raising this issue. We revised the manuscript accordingly (Page 5: "385").

3. 80% of the features had correlation >0.7 between duplicates, which did not seem exceptionally high. Was there something about the other 85 features that contributed to the lack of correlation?

To address the reviewer's question, the list of all 385 features that we now provide as an additional Supplementary Table (Supplementary Table III) also includes information for the correlation between biological replicates for each of the features.

Among the features that had a correlation <0.7 , many were averages of cellular moment features such as centre of mass coordinates in pixels, the angle between fitted elliptical axes of the object and x- and y-axes of the image, and similar (for a more comprehensive explanation of additional features, please see the 'computeFeatures' manual page of the R package EImage). For the data considered here, it is expected that such quantities are not correlated between replicates - indeed it is a signature of data quality, since such correlations would reflect problems such as cell-seeding biases.

These features are computed as part of a comprehensive feature set calculation by the EImage software; in other applications, the same features might indeed be informative (e.g. for single-cell or cell-cell interaction analyses). Here, instead of manually identifying them, we simply took the more objective, data-driven criterion of filtering them out via lack of replicate correlation.

4. Cells were treated for 2 days with 5 μ M of each compound, which seems long and high for many of these common bioactive compounds. I would imagine many toxic effects. Perhaps a shorter treatment would increase correlation? It might be worth discussing in the manuscript. For example, it is difficult to interpret the results of wortmannin used at 5 μ M, as it is hitting quite a number of targets at that concentration.

The reviewer is correct in that using a concentration of 5 μ M for all compounds and a single time point of 48 h is not ideal. It is likely that different time-points or the use of multiple concentrations would provide additional insights for specific compounds. However, given that resources are finite, such considerations always imply a trade-off between 'breadth' and 'depth', i.e. between experimental throughput and detail. For instance, Perlman et al. (2004) performed high-content screens of phenotypic effects using 100 drugs at 13 different concentrations. In contrast, we employed 1280 compounds at a single concentration. Our choice of a relatively high concentration was influenced by the consideration that surprise effects of relatively specific, non-toxic drugs would be more interesting than modulation of very toxic drugs.

We are aware of this limitation of our current experimental pipeline and have included a respective statement in the Discussion section: "*Several extensions of the phenotypic pharmacogenetic screening assay presented here will be desirable. For example, the resolution of the data would benefit from using multiple doses of each drug (Perlman et al., 2004)....*" (Page 16).

5. How are the phenoprints compared quantitatively? The impression from the paper is that they are visually similar by eye, but that cannot be sustainable across so many compound treatments.

We apologise that our previous text describing phenoprints might have been imprecise. Phenoprints themselves are not directly used to quantify (dis-)similarities between drug-induced phenotypes. Our intention for the phenoprints is to serve as an easily discernable visualization of the 20 features that we use as phenotypic fingerprints. To clarify this we have now altered the text: “...phenoprints served as compact visualizations of drug-induced phenotypes.” (Page 6).

To compare phenotypes quantitatively, correlations of feature vectors are used. This strategy has been employed to generate the plotted matrix shown in Supplementary Figure S12 and is conceptually similar to previously used approaches (Young et al, 2007, Gustafsdottir et al., 2013).

6. It is mentioned in the methods that serial dilutions of each compound were made, but there is not discussion of the effects of dose on phenotypes. Were the compounds tested in dose? If not, that point needs to be clarified.

For the analysis of drug combinations effects, we used concentration-kinetic measurements. For screening we solely tested a single concentration. A respective statement has now been included to clarify this point: “Prior to screening, we prepared serial dilutions of the LOPAC compound library (Sigma) in RPMI medium (Life Technologies) to provide a final stock concentration of 50 μM ...For screening, a single drug concentration of 5 μM was used.” (Page 18).

7. Images need to be larger, or include some insets of higher magnification. It is impossible to see what the authors are trying to convey.

We thank the reviewer for this helpful suggestion. We have now included higher magnification images to improve visualization (see below).

Revised Figure 2A.

8. Similarly, in Figure S2, it is not clear what the reader is supposed to see in this image. A control should be included for greater clarity.

We have now included arrows to highlight the occurrence of artefacts in Figure S2. As suggested by the reviewer, we have also included a control image for greater clarity (see below).

Revised Figure S2.

9. The lower number of chemical-genetic interactions based on cell number is confusing, as the other publications looking at chemical-genetic interactions using cell viability see quite a few interactions. Some discussion of the lower number here would be helpful.

The higher number of chemical-genetic interactions observed in other publications using cell viability as a readout is presumably based on 3 factors:

- i) Larger cell line panels have been screened. For example, Basu et al. employed ca. 250 cell lines and Barretina et al. used ca. 500 cell lines (Basu et al. 2013; Barretina et al., 2012).
- ii) These cell lines originated from different lineages and present very heterogeneous genetic backgrounds with multiple alterations thereby increasing the likelihood to observe gene-drug interactions (although it might be more difficult to pinpoint specific gene-drug interactions given the presence of potentially confounding co-alterations within the complex genetic backgrounds of cell lines).
- iii) Focused compound libraries that preferentially included drugs known to affect the viability of cancer cells were used in many published gene-drug interaction screens using human cell lines (e.g. Basu et al. 2013; Barretina et al., 2012), hence increasing the likelihood to observe chemical-genetic interactions that affect cell growth.

In our screen, we employed 12 isogenic colon cancer cell lines with specific genetic alterations and used a compound library not exclusively restricted to FDA-approved or novel anti-cancer agents. We have already included a comment on this in the previous manuscript version: *“We chose a drug library with 1280 pharmacologically active compounds affecting a broad spectrum of cellular processes and major drug target classes (Supplementary Table S1) in order to obtain a comprehensive view of phenotypic pharmacogenetic effects that should not exclusively be restricted to well-established anti-cancer agents.”*

We have now extended our previous comment in the Results section to read: *“This design*

choice contrasts with other studies that focused on oncology-related drugs.“ (Page 5).

We have additionally extended our comment in the Discussion section to address the reviewer’s concern: “...*Several extensions of the phenotypic pharmacogenetic screening assay presented here will be desirable. ... a broader set of genetic backgrounds could be used, an aim that now appears quite tractable with isogenic cell lines through use of CRISRP/Cas9 technology (Sander and Joung, 2014).*“ (Page 16).

10. In the proteasome inhibition assay, what is the effect of DMSO alone? Is it pegged at zero? This would help interpret the graph shown in Figure 6.

The reviewer is correct that DMSO control determines 0% proteasome inhibition. This is now stated in the legend to Fig 6: “*DMSO control determines 0% normalized proteasome inhibition.*” (Page 36).

For clarification, we have additionally extended the Material and Methods section: “...*Consequently, DMSO control determines 0% normalized proteasome inhibition.*” (Page 23).

Thank you again for submitting your work to Molecular Systems Biology. We are now globally satisfied with the modifications made and we will be able to accept your paper for publication pending the following points:

- We appreciate that you provide the Vignette of the PGPC package as PDF files. For long term archival purpose, it would also be good to include the source package including the vignette and the respective datasets as zip file ('Expanded View Computer Code'). Please confirm that the package will be made available on Bioconductor as soon as the paper is accepted (we could not yet find it with the search <http://www.bioconductor.org/help/search/index.html?q=PGPC/>).

- It is a great idea to provide the resource at <http://dedomena.embl.de/PGPC/> to browse through the data. It does not seem however that the site allows to access/download the full imaging dataset as such or the extracted features and statistics. The dataset should be made available to the community, either by depositing it to Dryad (datadryad.org) or a similar resource such as Biostudies (<http://www.ebi.ac.uk/biostudies/>). Please include a 'Data availability' section at the end of Materials & Methods to provide explicit links/identifiers to the software (PGPC package) and the datasets (raw imaging and extracted features).

- As you may have noticed, we recently replaced Supplementary Information by Expanded View (EV, see examples in <http://msb.embopress.org/content/11/6/812>). In this format, a limited number of Supplementary Figures (maximum 5) can be integrated in the article as EV figures that are interactively collapsible/expandable and will be typeset by the publisher. In this case, the figures should be cited as 'Figure EV1, Figure EV2' etc... in the text and their respective legends should be added to the main text after the legends of regular figures. The illustrations should be provided as separate files.

- For the figures that you do NOT wish to display as Expanded View figures items, they should be bundled together with their legends in a 'traditional' supplementary PDF, now called the *Appendix*. Appendix should start with a short Table of Content and the figures should be named and referred to in the main text as: "Appendix Figure S1, Appendix Figure S2" etc. See detailed instructions regarding expanded view here: <http://msb.embopress.org/authorguide#expandedview>.

- Additional Tables/Datasets should be labeled and referred to as Table (or Dataset) EV1 etc. Table/Dataset legends can be provided in a separate tab in case of .xls files. Alternatively, you can upload a .zip file containing the Table/Dataset file and a separate README .txt file with the legend/description.

We thank you for the positive comments on our revised manuscript and for accepting our paper for publication in Molecular Systems Biology. Following your instructions, we have added a *Data Availability* section to *Material and Methods* and provide detailed information how readers can access the analysis package PGPC. The R package is freely available to the scientific community at Bioconductor (<http://bioconductor.org/packages/PGPC>). Its vignette has been included in the *Expanded View Computer Code* section. Since the R package has a large size, one option may be to provide on your website only the vignette as PDF together with the above link to Bioconductor, instead of the whole file – but this is a decision the journal should make. We have initiated the storage of the complete image dataset (2 TB) in a public repository. There is no established database for such image-based screens, however, we are in contact with EMBL-EBI, who are currently

setting up the EMBL-EBI / BBSRC Image Data Repository (<http://idrdemo.openmicroscopy.org>). Due to the size of the data but more importantly their novel type and the experimental nature of the repository, the data transfer and repository-side quality control will likely take some more time. We suggest that this process should not delay the publication of this paper, since the images can be browsed at <http://dedomena.embl.de/PGPC> and we are happy to provide the data on hard disk drives to all interested readers in the meanwhile, until the EMBL-EBI repository is fully operational.

We have now also included three Expanded View Figures as separate files, as well as an Appendix including all supplementary Figures and Information. The main text has been modified accordingly. We thank you again for considering our work for publication in *Molecular Systems Biology*. Do not hesitate to contact us for further information related to our manuscript.